# Disinfecting Action of Gaseous Ozone on OXA-48-Producing *Klebsiella pneumoniae* Biofilm In Vitro

**DOI:** 10.3390/ijerph19106177

**Published:** 2022-05-19

**Authors:** Kaća Piletić, Bruno Kovač, Marko Perčić, Jure Žigon, Dalibor Broznić, Ljerka Karleuša, Sanja Lučić Blagojević, Martina Oder, Ivana Gobin

**Affiliations:** 1Department of Microbiology and Parasitology, Faculty of Medicine, University of Rijeka, 51000 Rijeka, Croatia; kaca.piletic@student.uniri.hr (K.P.); bruno.kovac@student.uniri.hr (B.K.); 2Faculty of Engineering & Centre for Micro- and Nanosciences and Technologies, University of Rijeka, 51000 Rijeka, Croatia; marko.percic@uniri.hr; 3Department of Wood Science and Technology, Biotechnical Faculty, University of Ljubljana, 1000 Ljubljana, Slovenia; jure.zigon@bf.uni-lj.si; 4Department of Medical Chemistry, Biochemistry and Clinical Chemistry, Faculty of Medicine, University of Rijeka, 51000 Rijeka, Croatia; dalibor.broznic@uniri.hr; 5Department of Physiology and Immunology, Faculty of Medicine, University of Rijeka, 51000 Rijeka, Croatia; ljerka.karleusa@uniri.hr; 6Faculty of Chemical Engineering and Technology, University of Zagreb, 10000 Zagreb, Croatia; slucic@fkit.hr; 7Department of Sanitary Engineering, Faculty of Health Sciences, University of Ljubljana, 1000 Ljubljana, Slovenia; martina.oder@zf.uni-lj.si

**Keywords:** biofilm, gaseous disinfection, hospital-acquired infections, *K. pneumoniae*, multidrug-resistant microorganisms, OXA-48 disinfection, ozone

## Abstract

*Klebsiella pneumoniae* is an emerging multidrug-resistant pathogen that can contaminate hospital surfaces in the form of a biofilm which is hard to remove with standard disinfectants. Because of biofilm resistance to conservative disinfectants, the application of new disinfection technologies is becoming more frequent. Ozone gas has antimicrobial activity but there is lack of data on its action against *K. pneumoniae* biofilm. The aim of this study was to investigate the effects and mechanisms of action of gaseous ozone on the OXA-48-procuding *K. pneumoniae* biofilm. A 24 h biofilm of *K. pneumoniae* formed on ceramic tiles was subsequently exposed to different concentrations of ozone during one and two hours to determine the optimal ozone concentration. Afterwards, the total bacteria count, total biomass and oxidative stress levels were monitored. A total of 25 ppm of gaseous ozone was determined to be optimal ozone concentration and caused reduction in total bacteria number in all strains of *K. pneumoniae* for 2.0 log_10_ CFU/cm^2^, followed by reduction in total biomass up to 88.15%. Reactive oxygen species levels significantly increased after the ozone treatment at 182% for the representative *K. pneumoniae* NCTC 13442 strain. Ozone gas in the concentration of 25 ppm caused significant biofilm reduction but did not completely eradicate the *K. pneumoniae* biofilm formed on ceramics. In conclusion, ozone gas has great potential to be used as an additional hygiene measure in joint combat against biofilm in hospital environments.

## 1. Introduction

One of the most important issues in modern medicine challenges are hospital acquired infections (HAI) caused by multidrug-resistant organisms (MDRO) [1]. These infections have a major impact on increased morbidity and mortality, hospital treatment-related complications and the overall cost of the treatment [2]. In the past decade, the number of HAIs caused by *Enterococcus faecium*, *Staphylococcus aureus*, *Klebsiella pneumoniae*, *Acinetobacter* spp., *Pseudomonas aeruginosa* and *Enterobacter* spp., also acronymically termed as the ESKAPE group, is on the rise and makes up about 2/3 of all infections, including HAIs [3,4,5]. *K. pneumoniae* is a prominent Gram-negative bacteria with notorious characteristics regarding its capabilities of forming biofilm, resistance to antibiotics and standard disinfectants [5,6]. *K. pneumoniae* is well known for its ability to produce different ranges of ß-lactamase enzymes capable of hydrolyzing ß-lactam rings in antibiotics such as penicillins, cephalosporins and carbapenems, and very often it is multidrug resistant [5,6,7,8,9]. Being a virulent opportunistic pathogen, it can cause serious infections in elderly and immunocompromised patients [10,11,12,13]. The major clinical concern nowadays is the overall presence of the carbapenem-resistant strain of *K. pneumoniae*, known as OXA-48-producing *K. pneumoniae*, which is slowly becoming more frequent-nosocomial strain in some European countries [14,15,16,17,18]. The main sources of contamination with *K. pneumoniae* in hospitals are in the gastrointestinal tract of both patients and staff, staff hands and the hospital environment indirectly and possibly in biofilm [10]. Biofilm is a highly structured community of microorganisms attached to a surface with features such as increased biocidal and antibiotic resistance as well as increased tolerance to desiccation [19,20]. About 90% of the biofilm structure is extracellular polymer substance (EPS), which has a protective role against environmental conditions, disinfectants and oxidative stress caused by external factors [19,20]. Although biofilm was historically connected mainly with moist environments, the literature shows that about 90% of hospital surfaces are contaminated with biofilm [19,20]. The potential impact of *K. pneumoniae* biofilm contamination in HAIs control is significant; therefore, it is a necessity to implement appropriate cleaning methods and eradication policies within a hospital system. Finding a suitable disinfection agent to eradicate or reduce *K. pneumoniae* biofilm from hospitals is challenging due to the increased resistance of biofilms to standard cleaning methods and biocidal active substances [19,20,21,22,23,24,25]. Furthermore, standard biocidal substances can be toxic for staff and the environment, and have negative effects on the materials after prolonged use [1,25,26,27]. Moreover, the use of conservative disinfectants in hospitals, especially in pandemic times, leaves a great amount of solid chemical waste, and is potential source of waste waters burdened with different chemicals [28]. Gaseous disinfection agents could be the answer to this hygiene control and environmental challenge. The application of gaseous disinfectants is proven to be effective because, in addition to the antimicrobial effect, gas can reach surfaces difficult to reach by conventional cleaning [29,30,31]. Ozone is triatomic allotropic oxygen modification and can be created when using high energy on molecular oxygen. It has proven antimicrobial properties, it is cheap to produce and, although toxic to humans in higher concentrations, ozone relatively rapidly dissociates back to oxygen and leaves no physical or chemical waste products [31,32,33,34,35,36,37]. There is a lack of scientific data on the antimicrobial effect and mechanism of action on *K. pneumoniae* biofilm. In this study, the efficacy and mechanisms of action of gaseous ozone on *K. pneumoniae* biofilm formed on ceramic tiles were explored.

## 2. Materials and Methods

### 2.1. Equipment

For the purposes of in vitro experiment, the test chamber made of transparent polystyrene sides and a coated plywood lid and bottom was crafted (Figure 1). The lid of the chamber was removable for easier access to the plates and had a hole for ozone input tube. Volume of the chamber was 125 L. The ozone generator used for this study was a portable model Mozon GPF 8008 provided by company Mozon d.o.o. The generator produces 5 g of ozone/air (O_3_/O_2_) mixture per 1 h. Ozone generated from the device was transferred to the chamber via rubber tube (6 mm diameter). Ozone concentration in the chamber was monitored continuously with a portable ozone detector model Keernuo GT901, China. Room temperature and humidity were monitored with a portable station Auriol 4-LD5531, Germany.

### 2.2. Characterization of Ceramic Tiles

Upper, smooth surface of mosaic ceramic tiles (2.5 × 2.5 cm) was used as ceramic surface for the biofilm formation. Tiles were mechanically brushed, washed with soap and water and then autoclaved for 15 min at 121 °C. After sterilization, tiles were characterized according to their physical properties of hydrophobicity and roughness. Hydrophobic/hydrophilic properties were measured using the OCA 20 goniometer (Data Physics Instruments GmbH) by measuring the contact angle using the sessile drop method (n = five). Volume of test liquid (water) was 2 µm. Measurements were made at 25 °C. The topology of the ceramic tile surface (n = five) (roughness) was analyzed with the Dimension Icon atomic force microscope (AFM) (Bruker, Billerica, MA, USA) using the tapping method. The tapping method was used to obtain a scanned tile surface using a SNL-10 type D silicon nitride bracket (Bruker) with a 2 nm radius silicon tip. The scan was performed with sizes of 2 and 5 µm^2^ with 512 scan lines, while each scan line had 512 data points collected per line. The obtained data were processed to obtain the values of the surface roughness parameters after the slope and bow correction using the proprietary Nanoscope Analysis software v1.5 (Bruker, Billerica, MA, USA).

### 2.3. Bacterial Strains

Bacterial strains used for the purposes of this study were standard strains of *K. pneumoniae* ATCC 700603 and *K. pneumoniae* NCTC 13442 obtained from the culture collection of the Department of Microbiology and Parasitology, University of Rijeka. Clinical isolates of OXA-48-producing *K. pneumoniae* strains: Kp strain 14, Kp strain 15, Kp strain 16, Kp strain 33 and Kp strain 34 obtained by the courtesy of General Hospital, Dr. Ivo Pedišić, Sisak were also used. OXA-48-producing *K. pneumoniae* isolates were determined by rapid in vitro diagnostic tests by the Coris BioConcept RESIST-5 O.O.K.N.V., model K-15R9. The bacteria were stored in 10% glycerol broth at −80 °C for later use.

### 2.4. Characterization of Bacterial Strains

Antimicrobial profile towards chosen antimicrobial drugs was determined for all tested strains of *K. pneumoniae*. Moreover, the characterization of their physical properties as hydrophilic/hydrophobic was conducted. 

#### 2.4.1. Antimicrobial Resistance Profile 

Antimicrobial resistance of all *K. pneumoniae* strains was determined by a standard disc diffusion method. Results were interpreted according to EUCAST Breakpoint Tables version 12.0 T. The minimal inhibitory concentration (MIC) for colistin was determined using characterization according to the manufacturer’s instructions (AB Biodisk, Solna, Sweden).

#### 2.4.2. Characterization of Surface Physical Properties

Hydrophilic/hydrophobic nature of bacterial strains was determined using the MATS (microbial adhesion to solvents) test developed by Bellon and Fontaine et al. [38]. Solvents used in this assay were chloroform, hexane and diethyl ether, each separately. Bacterial suspension in NaCl 0.15 mol L^−1^, approximately 10^8^ CFU/mL and the chosen solvent was shaken in a vortex for 2 min to form an emulsion. After 15 min incubation, the absorbance aqueous phase was measured at 400 nm. The percentage of bound cells to each solvent was calculated by the equation
(1)$$\%Adh=\left(1-\frac{A}{Ao}\right)\times 100$$
where *Ao* was the absorbance of the bacterial suspension before mixing and *A* was the absorbance after mixing [39].

### 2.5. Biofilm Formation on Ceramic Tiles

For the formation of biofilm, the smooth surface tiles were brushed, washed, sterilized and the biofilm was formed on the upper surface of the tiles. The method for biofilm formation was previously described by Ivanković T. et al. [40], with some modifications. Firstly, agar was prepared. The tiles were placed in a petri dish and covered with liquified agar (2% *v*/*v*), leaving the upper surface of the tiles free from agar. A small amount of bacterial culture was suspended in 5 mL of Muller Hinton broth (MHB) and left to incubate at 37 °C overnight. Overnight bacterial suspension was diluted to 10^5^ CFU/mL and poured over the upper surface of the ceramic tiles in agar in petri dishes. Inoculated petri dishes were then incubated for biofilm formation in an orbital shaker at 30–50 rpm for 24 h at 25 ± 2 °C.

### 2.6. Determination of Optimal Gaseous Ozone Concentration

The experiment was done to determine the optimal ozone concentration that causes significant reduction of the *K. pneumoniae* biofilm. To determine that concentration, 24-hour-old *K. pneumoniae* ATCC 700603 and *K. pneumoniae* NCTC 13442 biofilm was treated with different ozone concentrations during a 1 h exposure time (5 ppm, 10 ppm, 15 ppm, 25 ppm, 50 and 75 ppm). Ceramic tiles with formed biofilm in petri dishes were washed with sterile saline solution and dried out in a laboratory safety cabinet for 1 min. The open petri dishes containing the ceramic tiles with the 24-hour-old biofilm were then placed in various locations in the ozonation chamber and then treated with different ozone concentrations for 1 h. All tests were performed at a relative humidity 56–58% and temperature was 23 ± 2 °C. After the exposure time ran out, the ceramic tiles were carefully removed from the agar and washed out with a sterile saline solution, placed in sterile tube containing 10 mL of sterile saline solution and sonicated in an ultrasound bath (Bandelin-BactoSonic, Berlin, Germany) at 40 kHz for 1 min to enhance the release of the adhered cells from the tiles. Then, the samples were homogenized using a vortex to enhance further, and the final detachment of the remaining cells from the biofilm into the sterile saline solution. Then, an aliquot of 200 µL of solution was used to prepare ten-fold serial dilutions inoculated onto MH agar and incubated for 24 to 48 h at 35 ± 2 °C. The number of cultivable bacteria was determined, and the results expressed as CFU/cm^2^. Control exposures were performed with the same relative humidity, room temperature and exposure time, but without ozone. The experiment and controls were done in triplicate within each replicate experiment. Additionally, for doses 25, 50 and 75 ppm, a 2 h exposure time was tested.

### 2.7. Determination of Total Bacterial Number

The determined optimal ozone concentration that caused a significant reduction (25 ppm/1 h) was used on further tests on *K. pneumoniae* biofilm (both standard and clinical strains) according to the protocol described in Section 2.6. Control exposures were performed with the same relative humidity, room temperature and exposure time, but without ozone. The experiment and controls were done in triplicate within each replicate experiment. 

### 2.8. Determination of Cell Viability (Dead/Live Assay)

For the determination of bacterial cell viability, the LIVE/DEAD^®^BacLightTM Bacterial Viability Kit solution (Molecular Probes, Eugene, OR, USA) was used according to the manufacturer’s instructions. After the treatment, the tiles were carefully rinsed with a sterile saline solution. A working solution for fluorescent dyes was applied to the ceramic tiles and incubated in the dark for 15 min. The samples were rinsed with the sterile saline solution to remove excess dye. Fluorescence from the stained cells was observed using an FV300 confocal microscope (Olympus Optical Company, Tokyo, Japan) with a 40× LCPlanF lens. The emission maxima for these colors were about 480/500 nm for SYTO^®^ 9 color and 490/635 nm for PI color. Simultaneous two-channel recording was used to display the green and red fluorescence. Green fluorescence marked live bacterial cells while red marked dead bacterial cells. The obtained images were saved in TIFF format and further processed using ImageJ 1.47 (National Institute for Health, Bethesda, MA, USA). Samples and controls were done in triplicate. For controls, the untreated biofilms were stained.

### 2.9. Biomass Determination by Crystal-Violet Staining

The biomass production of *K. pneumoniae* on ceramic tiles was determined with crystal violet staining (CV). The treated ceramic tiles and controls were removed from agar, rinsed with sterile saline solution, fixated during 30 min on 80 °C in a dry heat sterilizer (ST-01/02, Instrumentaria, Zagreb, Croatia) and stained with 0.1% crystal violet for 30 min. Afterwards, the tiles were rinsed with 95% ethanol during 15 min with intense mixing, and the optical density was measured on a spectrophotometer (Eppendorf, Biophotometer, model #6131, Hamburg, Germany) at 600 nm. Stained ceramic tiles were also used later for digital microscopy. 

### 2.10. ATP Bioluminescence

A 3M Clean-TraceTM luminometer (3M, Saint Paul, MN, USA) was used to determine the efficiency of the ozone treatment. The entire surface of the treated plates and controls were wiped with a 3M Clean-TraceTM Surface ATP Test Swab UXL100, (3M, Saint Paul, MN, USA) swab preimmersed for 1 min in luciferin and luciferase reagent. After one minute, the wipe was placed in a 3M Clean-TraceTM luminometer, (3M, Saint Paul, MN, USA). The amount of light produced was read from the luminometer and was expressed in relative light units (RLUs) per tile.

### 2.11. Atomic Force Microscopy

Atomic force microscopy of the early 24 h biofilm of *K. pneumoniae* NCTC 13442 created on ceramic tiles was used to determine the biofilm topology before and after the ozone treatment, as described in Section 2.2. 

### 2.12. Digital Microscopy

The surfaces of the tiles with biofilms were first microscopically examined with the digital microscope (DM) DSX 1000 (Olympus, Tokyo, Japan) at 20× magnification. Analyses were performed at 3 different locations (1 mm × 1 mm) of each tile. The DM provided both two-dimensional and three-dimensional images of the analyzed spots. 

### 2.13. Scanning Electron Microscopy

To perform biofilm analysis by scanning electron microscopy, the ceramic tiles were analyzed using a Quanta 250 scanning electron microscope (Thermo Fisher Scientific, Waltham, MA, USA). Before the analyses, the ceramic plates were fixed with a 4% glutaraldehyde solution and dehydrated by double washing in buffer and in ethanol solutions (50% to 100%) for 20 min. A conductive gold layer was sputtered onto the samples prior to examination. Three selected areas on the tiles were analyzed at different magnifications. Images were acquired with the Everhart–Thornley detector in a high vacuum (0.0915 Pa) with an electron source voltage of 5.0 kV at a working distance of 10 mm and a spot size of 3.0 nm. 

### 2.14. Determination of Oxidative Stress

To determine the levels of oxidative stress in the selected bacterial strains treated with ozone, as well as in the controls, the level of reactive oxygen species (ROS) was measured using a method previously described by Rajneesh et al. [41]. Briefly, after washing the control and ozone-treated biofilm, 2 mol/L 2′,7′-dichlorodihydrofluorescein diacetate (DCFH-DA) solubilized in ethanol was added and incubated on a shaker (15 rpm) at room temperature in the dark for 1 h. After 1 h incubation, spectrophotometric analysis on the Fluoromax 3 (Horiba, Japan) (λexc = 485 nm, λem = 600 nm) was performed.

For the fluorescence microscopy DCFH-DA-stained bacteria (control and ozone-treated) were added on a clean glass slide and covered with a glass cover slip. Cells were visualized under a fluorescence microscope using an excitation wavelength of 488 nm, and emission was detected in the range of 500–600 nm. Oxidation of DCFH by ROS converts the molecule to 2′,7′-dichlorofluorescein (DCF), which emits green fluorescence.

The obtained images were saved in TIFF format and further processed with ImageJ 1.47 (National Institute for Health, Bethesda, MA, USA). A minimum of three images per term were analyzed. 

### 2.15. Statistical Analyses and Graphing

To evaluate effect of gaseous ozone using a method for total bacteria count determination, CV staining, ATP bioluminescence and for ROS level determination data were analyzed performing the Mann–Whitney test (*p* < 0.05). To evaluate effect of ozone between *K. pneumoniae* strains, a two-way analysis of variance (ANOVA) followed by the Kruskal–Wallis multiple comparison test (*p* < 0.05) was used. Graphing was done using Microsoft Excel, version 11.00 (Microsoft Home Office, Redmond, WA, USA. Pearson’s correlation coefficient was determined using the software TIBCO Statistica 14.0.0. (StatSoft Inc., Tulsa, OK, USA).

## 3. Results

### 3.1. Antimicrobial Resistance Profile of K. pneumoniae

*K. pneumoniae* strains were tested for antimicrobial resistance using the disc diffusion method. All strains of *K. pneumoniae* were multidrug resistant (Appendix A). The MIC was determined for colistin and Kp strain 14, Kp strain 15, Kp strain 16, Kp strain 34, Kp ATCC 700603 and Kp NCTC 13442. All strains were sensitive to colistin, while only Kp strain 33 was resistant. 

### 3.2. Characterization of Ceramic Tiles

Roughness and hydrophobic/hydrophilic properties of the surface were determined and expressed as root mean square (RMS). The calculated RMS value for the tested ceramic tiles was 8.41 ± 0.2 nm, indicating a smooth ceramic tile surface.

The contact angle measurements were conducted, and the measured contact angle of the tested samples was 33.42 ± 6.38°. Results from all measurements indicate the hydrophilic properties of the tile surfaces. 

### 3.3. Cell Surface Characterization of Bacterial Strains 

Cell surfaces of the tested *K. pneumoniae* strains presented more acidic properties and were hydrophilic regarding the affinity to hexane, with results ranging from 16.5 ± 0.00086% for Kp ATCC 700603 up to 19.47 ± 0.008% for the KpNCTC 13442 strain.

### 3.4. Optimal Concentration of Gaseous Ozone

During the determination of the optimal ozone concentration, the tested *K. pneumoniae* strains Kp NCTC 13442 and Kp ATCC700603 showed a slow progression in biofilm reduction after the ozone dosage increased (Figure 2A). The minimum effective dose of gaseous ozone was determined at 25 ppm, with a statistically significant difference in comparison to the control (*p* < 0.05) and a 99% inhibition rate (Appendix A). Interestingly, after 25 ppm of the ozone concentration, the increase in dose did not cause any further biofilm reduction. This can indicate the activation of the antioxidation mechanism in the biofilm or may be the result of disturbed ozone mass transfer to the biofilm due to certain physical–chemical properties. Moreover, during the two-hour exposure time, the biofilm reduction with different concentrations of gaseous ozone (25, 50 and 75 ppm) remained similar to the reduction obtained during the one-hour exposure (Figure 2B). Because of the observed effect, all further tests were done with the optimal ozone concentration (25 ppm) and 1 h exposure.

### 3.5. Total Bacteria Number

All tested strains of *K. pneumoniae* formed a biofilm on the ceramic tiles, and the number of bacteria varied from 6.0 log_10_ CFU/cm^2^ to 6.5 log_10_ CFU/cm^2^ (Figure 3). The ozone treatment with a concentration of 25 ppm/1 h significantly reduced the number of cultivable bacteria in biofilm from 2.0 log_10_ CFU/cm^2^ to 2.5 log_10_ CFU/cm^2^. There was a significant difference between *K. pneumoniae* ATCC 700603 (Kp ATCC 700603) in comparison to strains *K. pneumoniae* 14 (Kp strain 14), *K. pneumoniae* 15 (Kp strain 15), *K. pneumoniae* 33 (Kp strain 33) and *K. pneumoniae* 34 (Kp strain 34) (*p* = 0.0002, 0.0136, 0.0087 and 0.0325) as determined by Kruskal–Wallis multiple comparison test (*p* < 0.05). 

### 3.6. Effect of Gaseous Ozone on Biomass Production of K. pneumonia Strains

Total biomass reduction was observed for all tested strains of *K. pneumoniae* after ozone treatment with 25 ppm for 1 h exposure. The reduction was statistically significant in comparison to the control group for all tested strains of *K. pneumoniae* (*p* < 0.05) (Figure 4).

### 3.7. Effect of Gaseous Ozone on ATP Bioluminescence

Bacterial biomass measured using the ATP bioluminescence method showed a statistically significant (*p* < 0.05) reduction in biomass after treatment with ozone for all *K. pneumoniae* strains in comparison to the control group (Figure 5). Significant difference between different *K. pneumoniae* strains was observed (*p* < 0.05).

Pearson’s correlation coefficient (r) was used to determine the correlation between the CFU, CV and ATP bioluminescence method. The r value between CFU/cm^2^ and the ATP bioluminescence was 0.80; between CFU/cm^2^ and CV dying it was 0.84; between CV dying and ATP bioluminescence it was 0.73. All calculated values of Pearson’s coefficient indicate a strong correlation between the used variables, and the correlation between them was a positive correlation.

### 3.8. Biofilm Inhibition

Biofilm inhibition was determined using the following formula,
(2)$$\%\;inhibition=1-\frac{N_{treatment}}{N_{control}}\times 100$$
where $N_{treatment}$ represents the average of all CFU/cm^2^, ATP bioluminescence and CV measurements, after treatment with 25 ppm ozone during 1 h exposure, while $N_{control}$ represents the average of all measurements obtained from CFU/cm^2^, ATP bioluminescence and CV on the control group (Appendix A). All inhibition rates were separately calculated for each method used (CFU/cm^2^, ATP bioluminescence and CV).

Inhibition rates varied from 99.76% to 21.4% depending on the used method. 

### 3.9. Effect of Gaseous Ozone on Topology of K. pneumoniae Biofilm

Changes in biofilm topology were monitored with digital microscopy and atomic force microscopy. The exemplary two-dimensional and three-dimensional images of the control and ozone-treated (25 ppm, 1 h) tiles with the representative *K. pneumoniae* NCTC 13442 strain, as acquired with DM, are shown in Figure 6. The analysis showed that the relatively rough and uneven surfaces of the tiles influenced the presence and distribution of bacterial cells. These seemed to grow more easily and be more present in the hollow areas than in the flat areas or protrusions. The comparison of the areas covered with the crystal violet-stained bacteria revealed the effectiveness of the ozone treatment to reduce the presence of bacteria in biofilm.

Changes in biofilm topology after the ozone treatment with 25 ppm/1 h were observed with AFM (D, E, F), resulting in cell aggregation with empty areas without bacteria (Figure 7). Moreover, variations in the distribution chart that follows the distribution of peaks in the three-dimensional biofilm structure suggests that after the ozone treatment, the biofilm height is reduced from 725.4 nm to 158.5 nm and from 355.4 nm to 173.0 nm (E–F), respectively. 

### 3.10. Effect of Gaseous Ozone on Bacterial Cells

The treated representative strain *K. pneumoniae* NCTC 13442 biofilm showed some morphological changes, such as the recesses area in the biofilm’s three-dimensional topology (A–B) (Figure 8). Moreover, changes in the bacterial cell wall were observed. The ozone treatment damaged the bacterial cells, causing invagination to their membrane (Figure 8D). 

### 3.11. Effect of Gaseous Ozone on K. pneumoniae Viability

In a representative photograph of the biofilm of *K. pneumoniae* NCTC 13442 that was not exposed to ozone gas, the areas with dense clusters of viable cells in the biofilm were observed. Fluorescence intensity for viable cells (green) in the control group was 10.526 AU, while for dead cells (red), it was 0.772 AU (Figure 9, control). After 1 h treatment with 25 ppm of ozone (Figure 9, ozone treatment), the areas with biofilm destruction can been clearly seen. The number of viable cells decreased to 4.203 AU, and a higher number of dead cells (red) with a measured fluorescence intensity of 4.11 AU were present in these areas. 

### 3.12. Effect of Gaseous Ozone on ROS Production

A significant increase of 182% of ROS levels determined by the fluorescent dye was observed for the representative strain *K. pneumoniae* NCTC 13442 biofilm (Figure 10 left). Fluorescence microscopy images indicate oxidative stress in representative strain *K. pneumoniae* NCTC 13442 biofilm treated with 25 ppm of ozone for 1 h (Figure 10A,B).

## 4. Discussion

Ozone has proven antimicrobial properties, but there is lack of scientific data on its effect on the *K. pneumoniae* biofilm; therefore, the aim of this study was to investigate the antimicrobial effect and mechanism of action of gaseous ozone on the *K. pneumoniae* biofilm. Since bacterial adhesion to the surface is the most important stage in biofilm formation, and the hydrophobicity of the bacteria and the surface is a dominant factor in the adhesion process, the physical properties of hydrophobicity/hydrophilicity of ceramic tiles, as well as the hydrophobicity/hydrophilicity of all tested *K. pneumoniae* strains, was performed [42,43,44,45,46].

The hydrophobicity and roughness of the ceramic tiles have shown that the tile surfaces were smooth and hydrophilic, since all water contact angles were lower than 90°, which is in line with the previous findings of van Loosdrecht et al. [42] and Kwok and Neumann [43].

Furthermore, all tested strains of *K. pneumoniae* showed to be hydrophilic towards hexane. Given the hydrophilic properties of the ceramic tiles and the tested bacteria, all *K. pneumoniae* strains resulted in good biofilm formation on the tiles, but besides the surface properties, other conditions such as the presence of nutrients, EPS production, surface charge and the presence of fimbriae should also be considered [47].

To measure the effect of ozone gas on *K. pneumoniae* biofilm, different concentrations of ozone gas were used during 1 and 2 h so to determine the optimal concentration of ozone, which was found to be 25 ppm for 1 h exposure. The optimal ozone concentration for 1 h was used in all further experiments.

The effectiveness of ozone gas in the concentration of 25 ppm for the 1 h exposure time was measured by the determination of the total number of bacteria and the total biomass. A total of 25 ppm of ozone during a 1 h exposure time was able to significantly reduce the *K. pneumoniae* total bacteria count and the total biomass for all seven strains. The measured biofilm reduction rates varied from 99.76% to 22.14%, depending on the used method. The highest reduction was observed in the total viable bacteria counts, while the smallest reduction was detected with the crystal violet staining method. The observed difference can be explained by the different sensitivities of the chosen methods, as well as by the fact that ozone made cultivable bacteria change into the viable but nonculturable (VBNC) mode.

Pearson’s correlation coefficient (r) showed a positive correlation between the used methods of CFU, CV and ATP bioluminescence, which is contradictory to the results of Larson et al. [48], where there was no correlation between CFU and ATP bioluminescence, but in line with Tebbut et al., where a positive correlation was found [49].

Regarding ozone effectiveness, the results indicate that in a concentration of 25 ppm and at a 1 h exposure time, ozone gas did not completely eradicate the *K. pneumoniae* biofilm but did cause a significant reduction in comparison to the control group. In addition to the concentration of 25 ppm for one and two hours, higher ozone concentrations (50 and 75 ppm) were tested under the same conditions, but the results did not much differ from the results obtained with the optimal concentration and were statistically not significant. This is contrary to the data showing that higher ozone concentrations cause a higher reduction in bacteria [34,36,50,51]. This result can be the outcome of specific ozone kinetics, possibly due to the limitation by the rate of mass transfer of ozone beyond this concentration. A similar situation was described by Aydogan et al. on the *Bacillus subtilis* spores [52]. Another explanation may be that the biofilm exposed to oxidative stress enables the protective antioxidation actions in producing more EPS, or activating peroxide scavenging systems [22]. Further experiments on ozone kinetics and biofilm penetration with higher relative humidity are needed.

There are still a lot of gaps in knowledge about the gaseous ozone mechanism of action on biofilm. In general, it is known that ozone is a strong oxidizing agent that rapidly dissociates and generates reactive oxygen species (ROS). Since all the tested strains of *K. pneumoniae* showed a similar response to the ozone treatment, further investigation of the ozone mechanism of action was conducted on the representative *K. pneumoniae* NCTC 13442 strain biofilm. 

Gaseous ozone caused morphological changes in the biofilm topology and three-dimensional structure. Moreover, the ozone caused a damaging effect on the highest peaks in the three-dimensional structure, as well as the less dense areas within the microbial load in the biofilm. Furthermore, damage of the bacterial cell walls was observed with the invaginated parts in the membrane, but with no leakage or cell protrusions. These observations go in line with previous results obtained by Nicholas et al. [53]. 

One of the most important elements of cell viability is membrane integrity and permeability [54]. Using Dead/Live, the areas with biofilm destruction with a greater number of dead cells were detected, pointing out to the ozone oxidizing action on membrane permeability and, consequently, cell death. Similar results were previously described by Nagayoshi et al. [55].

The results regarding the oxidative action of ozone indicate that the ROS levels rise significantly after the ozone treatment with 25 ppm for 1 h in comparison to the control group. In addition to the mentioned measurements, fluorescence microscopy images were taken, indicating oxidative stress in *K. pneumoniae* bacterial cells in the biofilm treated with ozone. These results go in line with previous studies, and once again describe ozone gas as a strong oxidizing agent on *K. pneumoniae*, causing a state of oxidative stress in the cells [32,33,34,35,36,37]. Some authors cite that the oxidative stress in the biofilm structure can result in extracellular polymeric substance overproduction [22,56,57], which is contrary to the results from Panebianco et al. [58]. 

Ozone seems to be very effective against planktonic bacteria, which are susceptible to ozone action and are often significantly reduced or completely eradicated from the surfaces with smaller concentrations [2,31,34,36,37,50,51,59,60,61,62,63,64]. Some authors, like Sharma et al. [2], used 25 ppm of ozone on 15 different planktonic bacterial strains, and the reduction rates were from 3 to 4 log_10_ CFU/cm^2^. Same concentration was used by Moat et al. [30] to achieve 2.7–3 log_10_ CFU/cm^2^ with shorter exposure times, but the situation changes with the susceptibility of bacterial biofilm to ozone. Studies on gaseous ozone action against *K. pneumoniae* biofilm are scarce. There are studies available on gaseous ozone action against *L. monocytogenes, S. aureus, P. aeruginosa* and *E. faecalis* biofilms [51,53,58,65,66,67]. All these studies used different concentrations of ozone that varied from 10–50, even 1000 ppm during different exposure times, but all have in common the conclusion that ozone as a solitary biocidal agent fails to completely eradicate bacterial biofilm. These results once again support previous findings by Vickery et al., Almatroudi et al., Hassett et al., Costa et al. and Smith K. [1,4,19,68,69] that bacterial biofilm, when formed on inanimate surfaces, is almost impossible to completely remove using only one method of disinfection, regardless of the chosen disinfectant. Moreover, during the COVID-19 pandemic, and the extensive disinfection in COVID-19 wards, when the number of other MDR pathogens seemed to decrease, *K. pneumoniae* contamination was on the rise, possibly indicating the strong resistance of this bacteria to disinfectants [70,71].

Although there are some studies, like Doan et al. [72], indicating that the cost-effectiveness of ozone disinfection in comparison to chlorine-based disinfectants leans on the side of chlorine, ozone gas undoubtedly has numerous advantages when applying to hospital environments. Because of the gas physical properties, it can fulfill a whole room volume and disinfect surfaces that are hard to reach with a classical disinfectant. It is also very cheap to produce and easy-to-handle when using mobile ozone generators. During pandemic times, hospitals and other healthcare institutions are using significantly more disinfectants, which subsequently become solid/liquid waste and a later problem for proper management and disposal. Das A. K. et al. mentions that 3% of medical waste originates from chemicals [28]. In comparison to those standard disinfectants in plastic packaging, ozone dissociates to oxygen, therefore leaving no waste, and it is considered environmentally friendly [2,21,30]. Furthermore, during the year 2020, the European Commission published a chemicals strategy for sustainability, which is part of the Union’s zero pollution ambition, one of the key commitments of the European Green Deal; therefore, the transition to sustainable, zero waste disinfectants could be favorable [73]. When comparing ozone to the other gaseous disinfectants, the most usual comparison is done with the effectiveness of hydrogen peroxide. The majority of available data [26,27,29] refers to the fact that hydrogen peroxide is a bit more effective than ozone, but the best effect was observed with the synergistic effect of ozone and hydrogen peroxide.

Ozone efficacy depends on concentration, exposure time, relative humidity, room temperature, bacterial form (planktonic or biofilm) and biofilm maturity [36]. Some of these parameters can be a limiting factor in the practical application of ozone in hospital environments, as antibiofilm biocide. Moreover, ozone is toxic to humans and can cause respiratory irritation in small concentrations. It is also considered to be genotoxic. The Occupational Safety and Health Administrations (OSHA) has set permissible exposure limits of ozone to 0.1 ppm, so practical application needs to be handled with caution and carried out without human presence [36].

Though our results indicate that gaseous ozone is not sufficient to fully mitigate *K. pneumoniae* biofilm from ceramic surfaces, it causes morphological changes in the bacterial cell wall and oxidative stress in biofilm bacterial cells. The optimal ozone concentration in this study was found to be 25 ppm for 1 h of exposure time. Hence, gaseous ozone disinfection can be used as an additional hygiene measure for the mitigation of *K. pneumoniae* biofilm contamination. 

## 5. Conclusions

Ozone gas in concentration of 25 ppm for 1 h of exposure was found to be the optimal concentration in this study and caused a significant reduction of the number of viable bacteria and the total biomass but did not fully remove the *K. pneumoniae* biofilm. Morphological changes of the biofilm topology and cell wall damage was observed in the form of invaginations, as well as an increase of the intracellular level of reactive oxygen species (ROS). 

Since bacterial biofilm is impossible to remove with only one disinfection method, these results indicate that ozone has a great potential for ecologically sustainable hospital disinfection when used in combination with mechanical cleaning and in combination with other disinfectants. 

Moreover, the evident lack of scientific data on gaseous ozone against *K. pneumoniae* biofilm should encourage further investigations of ozone gas efficacy and the mechanism of action to control hospital-acquired infections caused by this pathogen. Investigations of the potential synergistic effect of ozone gas in combination with other biocidal substances as a potential biofilm control measure is recommended.

## Figures and Tables

**Figure 1 ijerph-19-06177-f001:**
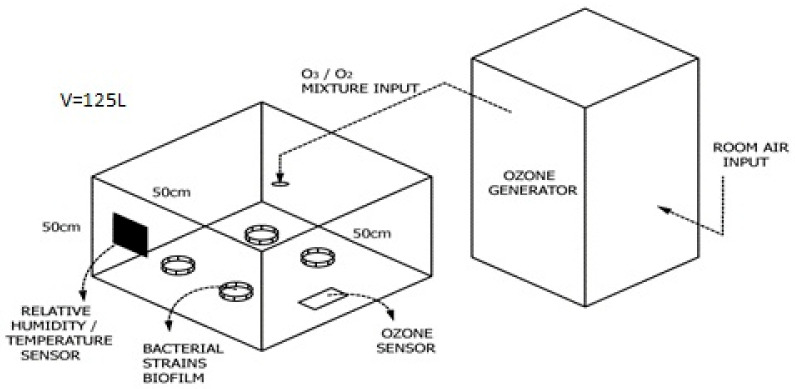
Ozonation in vitro model scheme.

**Figure 2 ijerph-19-06177-f002:**
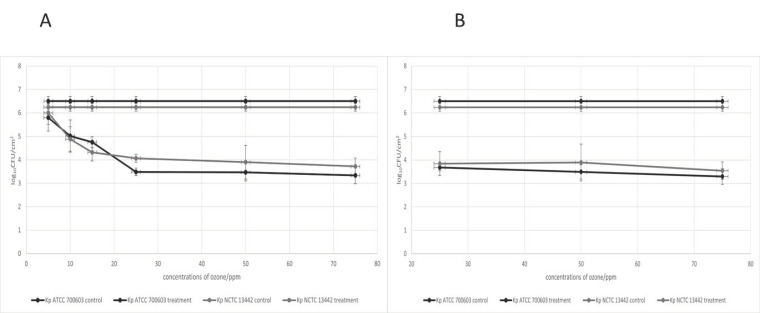
Antimicrobial effect on 24 h biofilm of *K. pneumoniae* NCTC 13442 (Kp NCTC 13442) and *K. pneumoniae* ATCC 700603 (Kp ATCC 700603) of different ozone concentrations (ppm) for 1 h exposure time (**A**) and 2 h exposure time (**B**). Results are presented with average (·) and standard deviation.

**Figure 3 ijerph-19-06177-f003:**
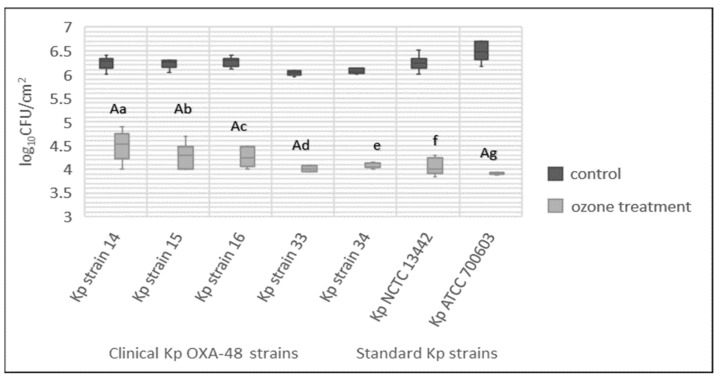
Total bacteria count expressed as log10 CFU/cm^2^ of 7 tested strains of *Klebsiella pneumoniae* (Kp). Results shown with median value (-) and minimum and maximum value. The lowercase letters, a–g, express the statistically significant difference between the treated and control group for the 7 tested strains (Mann–Whitney U test, *p* < 0.05). The capital letter A marks the statistically significant difference between Kp ATCC 700603 and OXA-48-producing strains Kp strain 14, Kp strain 15, Kp strain 16 and Kp strain 33, and vice versa (Kruskal–Wallis test, *p* < 0.05).

**Figure 4 ijerph-19-06177-f004:**
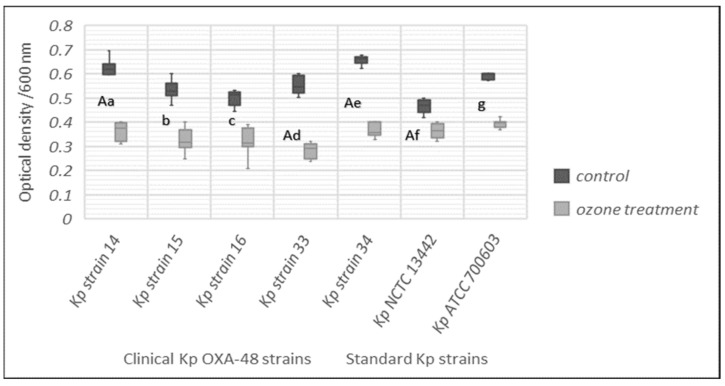
Biomass determination (CV staining) for control and treated *K. pneumoniae* strains (Kp) after treatment of 24 h biofilm with 25 ppm of ozone for 1 h. Results shown with median value (-) and minimum and maximum value. Lowercase letters, a–g, express the statistically significant difference between the treated and control group for the 7 tested strains (Mann–Whitney U test, *p* < 0.05). The capital letter A marks the statistically significant difference between strains Kp ATCC 700603, Kp strain 14, Kp strain 33, Kp strain 34 and Kp NCTC 13442, and vice versa (Kruskal–Wallis test, *p* < 0.05).

**Figure 5 ijerph-19-06177-f005:**
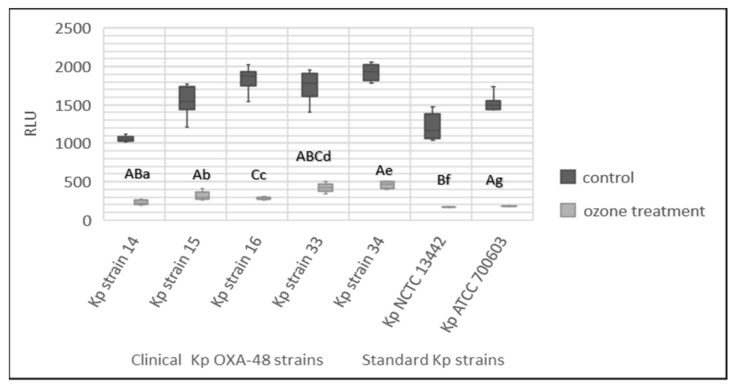
ATP bioluminescence method for control and treated *K. pneumoniae* strains after treatment of 24 h biofilm with 25 ppm of ozone for 1 h. Lowercase letters, a–g, express the statistically significant difference between the treated and control group for the 7 tested strains (Mann–Whitney U test, *p* < 0.05). The capital letter A marks the statistically significant difference between Kp NCTC 13442 and strains Kp strain 14, Kp strain 15, Kp strain 33 and Kp strain 34, and the capital letter B marks the significant difference between Kp strain 14, Kp strain 33 and Kp NCTC 13442, while C marks the statistically significant difference between Kp strain 16 and Kp strain 33 (Kruskal–Wallis test, *p* < 0.05).

**Figure 6 ijerph-19-06177-f006:**
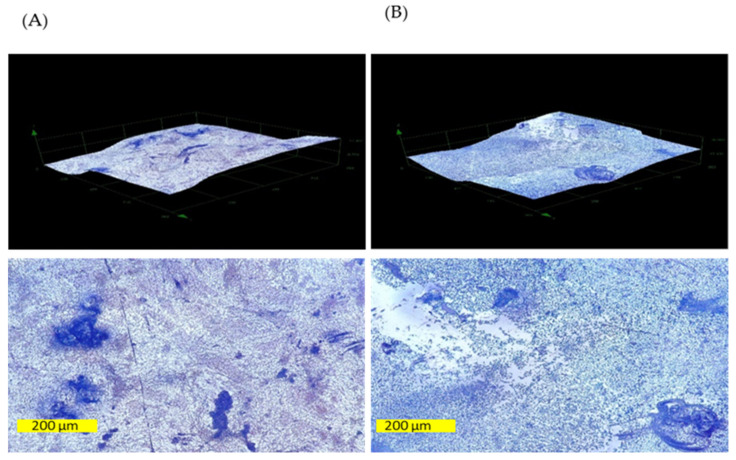
The three-dimensional (**above**) and two-dimensional (**below**) images of ceramic tiles surfaces inoculated with representative strain *K. pneumoniae* NCTC 13442 biofilm: control sample (**A**) and ozone-treated sample (**B**). The dark blue stains present the bacteria stained with crystal violet.

**Figure 7 ijerph-19-06177-f007:**
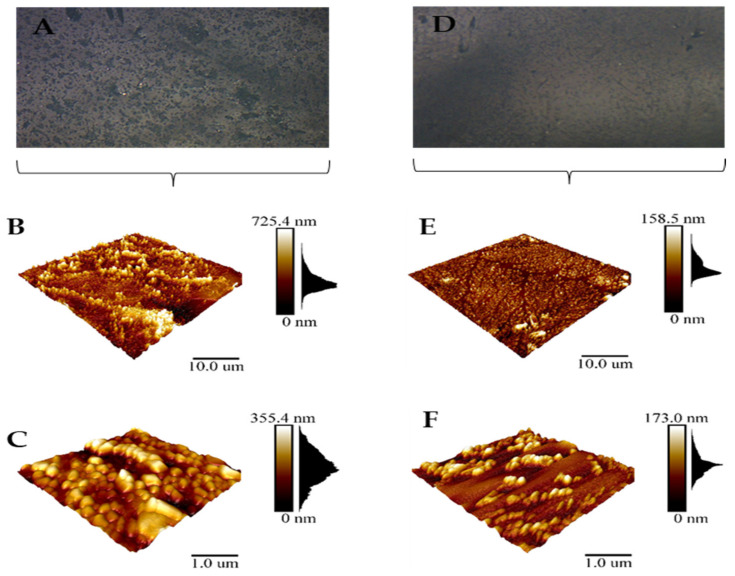
Atomic force microscopy of representative strain *K. pneumoniae* NCTC 13442 biofilm on ceramic tiles. Topology of biofilm created after 24 h (**A**–**C**) and topology of biofilm treated with 25 ppm of ozone for 1 h (**D**–**F**). Scanning area for (**B**,**E**) is 10 µm × 10 µm and for (**C**,**F**) is 1 µm × 1 µm.

**Figure 8 ijerph-19-06177-f008:**
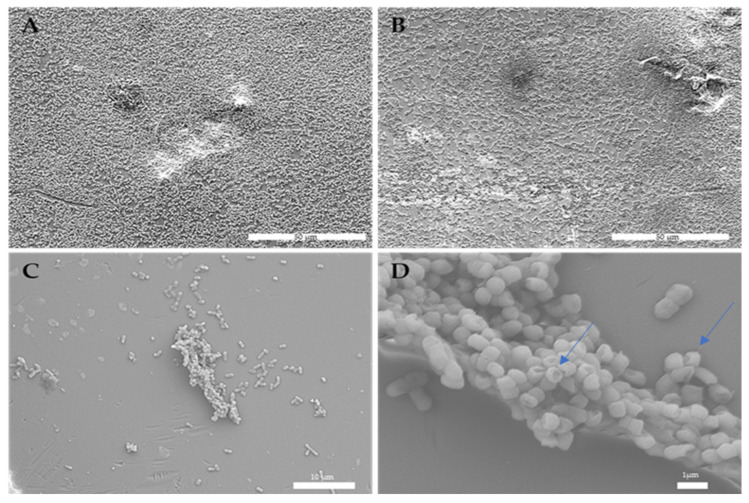
Morphological changes in representative strain *K. pneumoniae* NCTC 13442 biofilm biomass and bacterial cells after ozone treatment. Recess areas in the biofilm topology were observed (**A**–**C**), as well as bacterial cell surface alteration (**D**) (arrowhead).

**Figure 9 ijerph-19-06177-f009:**
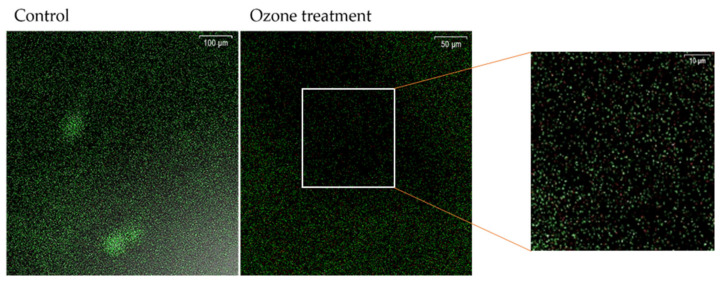
Viability of the representative strain *K. pneumoniae* NCTC 13442 cells in biofilm. Viability of nontreated control group, and 25 ppm of ozone/1h treated group using Dead/Live staining. Green fluorescence is representing viable cells with intact membrane, and red fluorescence indicates dead cells with permeable membrane.

**Figure 10 ijerph-19-06177-f010:**
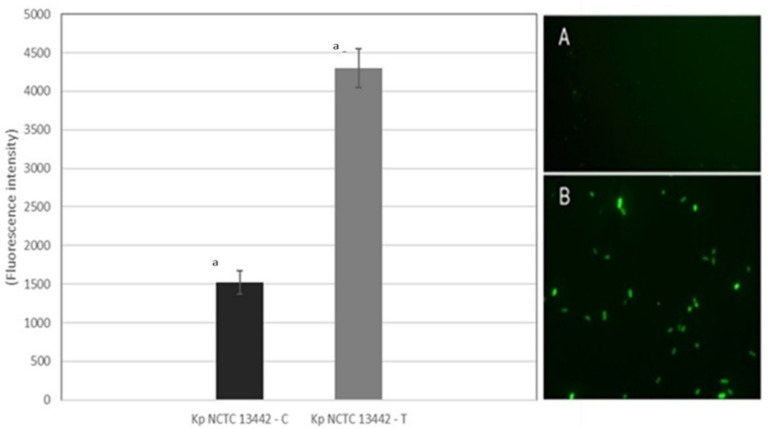
Measured fluorescence intensity for control and treated representative strain *K. pneumoniae* NCTC 13442 biofilm. Results shown with median, minimum and maximum values. The lowercase letter a marks the statistically significant difference between the control and treated group of *K. pneumoniae* NCTC 13442 strain. On the right are fluorescence microscopy images magnified 1000× of the control (**A**) and ozone-treated (**B**) representative strain *K. pneumoniae* NCTC 13442 biofilm. Excitation wavelength was 488 nm and emission were detected in the range of 500–600 nm.

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
