# Peer review of "Disinfecting Action of Gaseous Ozone on OXA-48-Producing Klebsiella pneumoniae Biofilm In Vitro"

_ijerph, 2022, doi:10.3390/ijerph19106177_

Round 1
Reviewer 1 Report
Disinfecting action of gaseous ozone on OXA-48-producing Klebsiella pneumonia biofilm in vitro
For remove the risk of drug-resistant Klebsiella pneumonia, authors used ozone to inactivate the biofilm of K. pneumonia. The results showed ozone was a potential disinfectant to remove the biofilm of K. pneumonia but not totally eradiated. The results are scientific significance. However, authors did not describe why used ceramic tiles as the testing material. There are many types of ceramic tiles which could be categorized into two kinds, hydrophilic and hydrophobic, for cleaning reason. In this study, author only use the hydrophilic kind and did not explain why only use the hydrophilic kind. In addition, only one treatment time period, 1 h, was tested. Since 25 ppm was the starting effective dose, why authors did not test higher concentrations of ozone or longer treatment time period to achieve better reduction? For practical used, most treatment time period of fumigation is longer than one hour. For scientific merit, it is good to know the dynamic correlation of treatment time and antibacterial effect. Hence, expecting authors offer more data of different concentration and treatment time.
- Introduction: many practices are used for hospital disinfection, including fumigation by gaseous disinfectants. Recommend authors describe the current practices and offer reasons why ozone is better than current ones.
- Line 84: why chose ceramic as the testing material?
- Line 88: revise to “the test chamber “
- Line 101: Were ceramic tiles glazed? Most ceramic tiles have two sides, one is smooth and the other is rough. Which side did authors use?
- Line 148: agar solution? This confuse me. Additionally, 2% agar should be solid not liquid. Were the tiles on the top of 2% agar no tin the agar?
- Line 143: was 24-h incubation enough to form biofilm?
- Line 149: there is no description for the preparation of overnight culture.
- Line 158: suggest to revise as ozone concentrations.
- Line 160: What is the digester?
- Line 167: “the samples were homogenized” How did authors homogenize the ceramic tiles.
- Section 2.14: The indication of green and red fluorescence was not described by authors.
- Section 3.5: did authors used ozone concentration above 25 ppm? Reduction of 2 log was not very effective and apparently higher concentrations were needed to achieve safer results.
- Section 3.8: Author did not explain the calculation of “??????????. Did all three components, CFU/cm2, ATP bioluminescence and CV measurements, contribute equally? It should be a description in the material and methods.
- Figure 9: only one letter “a” was marked for statistically significance.
- Line 415: Results indicated ceramic tiles used for this study were hydrophilic. These confuse me since most ceramic tiles on wall are hydrophobic and water-proof. This is why ceramic tiles are widely used in bathrooms and toilets. If authors use the rough side of tile, it is against the daily practice since the exposed side of tiles are smooth.
- Did authors test other treatment time? It would be a more complete study to have different treatment time periods.
- Line 480: the reference numbers are better immediately after the cited references.
- Line 487: Doan et al. (reference number). It is easier to read.
- Line 4874-504: there are many fumigants used for disinfection now. However, authors did not compare these fumigants with ozone. In addition, authors did not mention the toxic effect of ozone to human.
- Line 495: Kumar Das, Is Kumar a first name?
Author Response
Response to Reviewer #1 of the manuscript ID: ijerph-1695574 entitled: ″Disinfecting action of gaseous ozone on OXA-48-producing Klebsiella pneumoniae biofilm in vitro” by Kaća Piletić, Bruno Kovač, Marko Perčić, Jure Žigon, Dalibor Broznić, Ljerka Karleuša, Sanja Lučić Blagojević, Martina Oder and Ivana Gobin
General comment: For remove the risk of drug-resistant Klebsiella pneumonia, authors used ozone to inactivate the biofilm of K. pneumonia. The results showed ozone was a potential disinfectant to remove the biofilm of K. pneumonia but not totally eradiated. The results are scientific significance. However, authors did not describe why used ceramic tiles as the testing material. There are many types of ceramic tiles which could be categorized into two kinds, hydrophilic and hydrophobic, for cleaning reason. In this study, author only use the hydrophilic kind and did not explain why only use the hydrophilic kind. In addition, only one treatment time period, 1 h, was tested. Since 25 ppm was the starting effective dose, why authors did not test higher concentrations of ozone or longer treatment time period to achieve better reduction? For practical used, most treatment time period of fumigation is longer than one hour. For scientific merit, it is good to know the dynamic correlation of treatment time and antibacterial effect. Hence, expecting authors offer more data of different concentration and treatment time.
General response: The Authors would like to thank Reviewer #1 for her/his valuable comments on our manuscript. We have responded to all of the comments and revised the paper accordingly. Detailed responses to each comment are provided below. To provide further detail on reviewer general comment on usage of ceramic tiles we would like to emphasise that authors have used ceramic tiles because they are widely present in hospital environment and data on biofilm formation and ozone efficacy on biofilm formed on ceramics is very scarce. Authors would also like to emphasise that indeed results with contact angle measurement showed that smooth/upper side of ceramic tiles was hydrophilic. Authors have chosen that type of tiles because they correlated with hydrophilic nature of tested bacterial surface. Certain authors Like Zeraik et al. claim that bacterial adhesion will be better and stronger when hydrophilic bacteria adhere on hydrophilic surface and vice versa. Authors wished to be sure in good bacterial adhesion and subsequent good biofilm formation on the tiles, therefore hydrophilic tiles were chosen. Regarding ozone concentration of 25 ppm as starting effective dose and exposure time at 1 hour, authors wish to explain that 25 ppm of ozone was chosen as optimal concentration due to the data obtained from numerous in vitro experiments. Authors did indeed conducted experiments with higher ozone concentrations (50 and 75 ppm) during 1 and 2 hour exposure time, but obtained data showed no statistical significance between bacterial reduction and 25 ppm dose. To further explain given results, higher concentrations of ozone (50 and 75 ppm) and exposure time (2 hours) caused similar reduction as 25 ppm or very small, statistically unsignificant increase in the number of dead bacterial cells in biofilm. Hence, authors have decided to continue to show ozone efficacy on this optimal concentration of 25 ppm and 1 hour exposure time.
Comment 1: Line 84: why chose ceramic as the testing material?
Response: The authors would like to thank Reviewer#1 for her/his valuable comments on our manuscript. Ceramic tiles were used because they are widely present in hospital environment and can pose risk for biofilm formation. Furthermore, data on biofilm formation and ozone efficacy on biofilm formed on ceramics is very scarce.
Comment 2: Line 88: revise to “the test chamber “
Response: Line 88 was revised to “the test chamber”.
Comment 3: Line 101: Were ceramic tiles glazed? Most ceramic tiles have two sides, one is smooth and the other is rough. Which side did authors use?
Response: The authors wish to claim that glazed ceramic tiles were used for the purposes of the experiment. All experiments were done on the upper, glazed side of ceramic tiles as this was the side on which bacteria formed biofilm on as well as the side which is available for bacterial adhesion when ceramic tiles are glued onto the walls of hospital rooms. Manuscript was revised in a way that sentence in line 101 was changed to “Upper, smooth surface of mosaic ceramic tiles (2.5 × 2.5 cm) was used as ceramic surface for the biofilm formation.”.
Comment 4: Line 148: agar solution? This confuse me. Additionally, 2% agar should be solid not liquid. Were the tiles on the top of 2% agar no tin the agar?
Response: The authors wish to acknowledge that “agar solution” is error in phrasing meaning that liquified agar was poured onto petri dishes. Subsequently it solidifies when cooling. Manuscript was changed in a way that word “solution” was deleted and word “liquified” was added.
Comment 5: Line 143: was 24-h incubation enough to form biofilm?
Response: The authors wish to thank reviewer 1 for his/her comment. 24 hours was enough time to form biofilm containing around 106CFU/cm2 of bacterial cells.
Comment 6: Line 149: there is no description for the preparation of overnight culture.
Response: Authors would like to apologise for the overlook of the description of the preparation of overnight culture. Manuscript was revised in a way that description of the preparation of overnight culture was added.
Comment 7: Line 158: suggest to revise as ozone concentrations.
Response: Authors would like to thank reviewer 1 for his/her valuable comment. As mentioned before, authors did use numerous concentrations in determination of minimum effective dose of gaseous ozone (10 ppm, 15 ppm, 25 ppm, 50 ppm and 75 ppm) but chose to depict and publish data only on the dose of 25 ppm which was minimum effective dose for statistically significant reduction of the biofilm. As seen in supplement table 2. this dosage caused inhibition of 99% of viable bacteria, when expressed as inhibition rate, which was significant inhibition rate (likewise many conventional biocidal products available on the market). Reasons why authors chose not to publish data on higher concentrations was because these data showed very similar results as 25 ppm or statistically insignificant higher reduction indicating that there are certain limitations in gaseous disinfectant absorption in biofilm matrix or that bacterial cells in biofilm activate certain protective mechanism. However, this matter needs further investigation to determine why higher concentrations don’t cause higher reduction when using gaseous ozone on K. pneumoniae biofilm in vitro.
Comment 8: Line 160: What is the digester?
Response: The digester is laboratory/chemical safe chamber with individual, separable air and ventilation control that pose safety environment both for researcher protection from chemical evaporation. Manuscript was revised in a way that word “digester” was replaced with “laboratory safety cabinet”.
Comment 9: Line 167: “the samples were homogenized” How did authors homogenize the ceramic tiles.
Response: Authors wish to say that the method for growing and enumerating biofilm formed on ceramic tiles was first described by Ivanković T. et al. Authors have used the method with some modifications. In the line 167 “the samples were homogenized” was meant to mean that after the ozone treatment, ceramic tiles from were carefully removed from agar and washed out with sterile saline solution, placed in sterile tube containing 10 mL of sterile saline and sonicated at ultrasound bath (Bandelin-BactoSonic, Germany) at 40 kHz for 1 minute to enhance release of the adhered cells from the tiles. Afterwards, sonicated sterile saline with tiles was homogenized during 1 minute on vortex to enhance further and final detachment of remaining cells from the biofilm into the sterile saline. Then 200 µL of this biofilm enriched sterile saline was used to make ten-fold serial dilutions. Manuscript was changed in a way that word “the samples were homogenized using vortex and aliquot of 200 µL of solution was used to prepare ten-fold serial dilutions”.
Comment 10: Section 2.14: The indication of green and red fluorescence was not described by authors.
Response: Authors wish to apologize for overlooking this matter. Manuscript was changed so that green fluorescence was described in section 2.14. by adding sentence “Oxidation of DCFH by ROS converts the molecule to 2’,7’-dichlorofluorescein (DCF), which emits green fluorescence”. Also, indication for green and red fluorescence was added in section 2.8. – Dead/Live staining.
Comment 11: Section 3.5: did authors used ozone concentration above 25 ppm? Reduction of 2 log was not very effective and apparently higher concentrations were needed to achieve safer results.
Response: As mentioned before, authors did use numerous concentrations in determination of minimum effective dose of gaseous ozone (10 ppm, 15 ppm, 25 ppm, 50 ppm and 75 ppm) but chose to depict and publish data only on the dose of 25 ppm which was minimum effective dose for statistically significant reduction of the biofilm. As seen in supplement table 2. this dosage caused inhibition of 99% of viable bacteria, when expressed as inhibition rate, which was significant inhibition rate (likewise many conventional biocidal products available on the market). Reasons why authors chose not to publish data on higher concentrations was because these data showed very similar results as 25 ppm or statistically insignificant higher reduction indicating that there are certain limitations in gaseous disinfectant absorption in biofilm matrix or that bacterial cells in biofilm activate certain protective mechanism. However, this matter needs further investigation to determine why higher concentrations don’t cause higher reduction when using gaseous ozone on K. pneumoniae biofilm in vitro. Biofilm is known to be challenging to completely remove from inanimate surface with only one disinfectant, regardless of the method used and numerous studies claim persistence of MDRO pathogens biofilms in hospital dry surfaces after terminal disinfection. Majority hospital cleaning protocols and intensive preventive and control measures regarding hospital infections propose multiple sanitation methods always combined with mechanical cleaning. In this study, authors claimed that ozone gas has good potential to be used in hospitals when used in combination because it exhibits strong oxidative action on bacterial cells in biofilm and causes morphological changes in bacteria which puts them in state of oxidative stress and makes them susceptible to action of usually used biocides. Authors think that ozone gas can be successfully used in hospitals to treat K. pneumoniae biofilm contamination, but having in mind biofilm resistance characteristics, always in combination with mechanical cleaning and other disinfectant.
Comment 12: Section 3.8: Author did not explain the calculation of “??????????. Did all three components, CFU/cm2, ATP bioluminescence and CV measurements, contribute equally? It should be a description in the material and methods.
Response: Authors wish to thank for the comment. Ntreatment stands for average number of all measurements for CFU/cm2, for ATP bioluminescence and for CV measurements separately for each method, after the treatment with 25 ppm ozone during 1 hour exposure. Manuscript was revised in a way that sentence “All inhibition rates were separately calculated for each method used (CFU/cm2, ATP bioluminescence and CV).” was added to the section 3.8.
Comment 13: Figure 9: only one letter “a” was marked for statistically significance.
Response: Only one letter “a” marks statistical significance because Figure 9. shows results only on representative strain of representative strain K. pneumoniae NCTC 13442 biofilm, so letter “a” represents statistically significant difference between representative strain K. pneumoniae NCTC 13442 and control strain. As authors have claimed in the manuscript, since all tested strains of K. pneumoniae showed a similar response to ozone treatment, further investigation of ozone mechanism of action was conducted on representative K. pneumoniae NCTC 13442 strain biofilm.
Comment 14: Line 415: Results indicated ceramic tiles used for this study were hydrophilic. These confuse me since most ceramic tiles on wall are hydrophobic and water-proof. This is why ceramic tiles are widely used in bathrooms and toilets. If authors use the rough side of tile, it is against the daily practice since the exposed side of tiles are smooth.
Response: Authors wish to thank reviewer 1 for his/her comment. Results from the contact angle measurements indeed showed that smooth, glazed, upper side of the surface was hydrophilic. Contact angle measurements were conducted, and measured contact angle of tested samples was 33.42 ± 6.38°. Since all measured contact angles were way below 90°, they can be characterized as hydrophilic. Authors wish to emphasize that these results were obtained from 12 measurements. Furthermore, authors have chosen that type of tiles because they correlated with hydrophilic nature of bacterial surface of K. pneumoniae strains which were hydrophilic. It was confirmed that there was good and strong adherence of K. pneumoiae to chosen ceramic surface, because there was 106 C oFU/cm2 of formed biofilm onto tiles.
Comment 15: Did authors test other treatment time? It would be a more complete study to have different treatment time periods.
Response: Authors did conducted experiments with 2 hour exposure time also, but obtained data showed no statistical significance between bacterial reduction and longer exposure time, maybe due to the degradation of ozone and its reaction with organic matter in the chamber. Given the mentioned facts, authors chose to use 1 hour as optimal exposure time, given in mind the practicalities in potential fumigation usage.
Comment 16: Line 480: the reference numbers are better immediately after the cited references.
Response: Manuscript was changed in a way that the reference numbers are put immediately after the cited references.
Comment 17: Line 487: Doan et al. (reference number). It is easier to read.
Response: Line 487 was changed in a way that reference number was placed immediately after Doan et al.
Comment 18: Line 4874-504: there are many fumigants used for disinfection now. However, authors did not compare these fumigants with ozone. In addition, authors did not mention the toxic effect of ozone to human.
Response: Authors wish to thank reviewer 1 for his/her valuable comment. Authors have added lines about comparison of effectiveness of ozone with hydrogen peroxide, as well as the lines about ozone toxicity to humans.
Comment 19: Line 495: Kumar Das, Is Kumar a first name?
Response: Kumar in line 495 is middle name of the author. Manuscript was revised so that word Kumar was deleted, only last name was left as reference.

Reviewer 2 Report
In this manuscript the Authors reported the effect of ozone on a K. pneumoniae biofilm. As already recorded by the authors, the gaeous mixture of ozone is one of the most important antimicrobial agent, and its efficacy is probably due to an increase in oxidative stress on the plasma membrane of the germ and the consequent cellular wall disruption.
Authors concluded that ozone , in concentration of 25 ppm, reduces but not eradicate K. pneumoniae biofilm.
The first question is:
why didn't the Authors use incremental doses of ozone? Can the Authors be sure that using higher doses of gaseous ozone would not have had a complete eradication of K. Pneumoniae?
In the manuscript is reported that various factors ( temperature, time, humidity) are important during the K.Pneumoniae biofilm eradication.
Since no one should be present during the disinfection of the room, is it no possible to predict alternative combinations of variables to achieve eradications?
Are ceramic tiles composed exclusively in the way illustrated by Authors in the experimental procedure? How important is the different composition of the tiles ( and the surface) on the gas ability to eradicate the biofilm?
Authors should spend more time to discuss the articles in which it is clearly shown that ozone completely destroy the germs at different doses and their selection of experimental variables ( especially the fixed dosage 25 ppm).
Author Response
Response to Reviewer #2 of the manuscript ID: ijerph-1695574 entitled: ″Disinfecting action of gaseous ozone on OXA-48-producing Klebsiella pneumoniae biofilm in vitro” by Kaća Piletić, Bruno Kovač, Marko Perčić, Jure Žigon, Dalibor Broznić, Ljerka Karleuša, Sanja Lučić Blagojević, Martina Oder and Ivana Gobin
General comment: In this manuscript the Authors reported the effect of ozone on a K. pneumoniae biofilm. As already recorded by the authors, the gaseous mixture of ozone is one of the most important antimicrobial agent, and its efficacy is probably due to an increase in oxidative stress on the plasma membrane of the germ and the consequent cellular wall disruption. Authors concluded that ozone , in concentration of 25 ppm, reduces but not eradicate K. pneumoniae biofilm.
General response: The Authors would like to thank Reviewer #2 for her/his valuable comments on our manuscript. We have responded to all of the comments and revised the paper accordingly. Detailed responses to each comment are provided below.
Comment 1: why didn't the Authors use incremental doses of ozone? Can the Authors be sure that using higher doses of gaseous ozone would not have had a complete eradication of K. Pneumoniae?
Response: Authors would like to thank reviewer #2 for his/her valuable comment. Authors did use higher dosage of gaseous ozone (50 ppm and 75 ppm) as well as numerous concentrations in determination of minimum effective dose of gaseous ozone (5 ppm, 10 ppm, 15 ppm, 25 ppm, 50 ppm and 75 ppm) but chose to depict and publish data only on the dose of 25 ppm which was minimum effective dose for statistically significant reduction of the biofilm. As seen in supplement table 2. of the manuscript this dosage caused inhibition of 99% of viable bacteria, when expressed as inhibition rate, which was significant inhibition rate (likewise many conventional biocidal products available on the market). Reasons why authors chose not to publish data on higher concentrations was because these data showed very similar results as 25 ppm or statistically insignificant higher reduction indicating that there are certain limitations in gaseous disinfectant absorption in biofilm matrix or that bacterial cells in biofilm activate certain protective mechanism. However, this matter needs further investigation to determine why higher concentrations don’t cause higher reduction when using gaseous ozone on K. pneumoniae biofilm in vitro.
Comment 2: In the manuscript is reported that various factors (temperature, time, humidity) are important during the K.Pneumoniae biofilm eradication.
Response: Authors would like to thank reviewer 2 for his/her valuable comment. All of the mentioned factors (temperature, time, humidity) were constantly measured during the experiments. During our measurements, authors chose to run ozone treatments with room temperature and humidity to better reproduce hospital room conditions.
Comment 3: Since no one should be present during the disinfection of the room, is it no possible to predict alternative combinations of variables to achieve eradications?
Response: During measurements, authors chose to run ozone treatments in vitro with room temperature (23°C) and humidity (50-60%) to better reproduce hospital environment conditions. Changing the variables of humidity and temperature would require changing the described methodology of the experiments.
Comment 4: Are ceramic tiles composed exclusively in the way illustrated by Authors in the experimental procedure? How important is the different composition of the tiles (and the surface) on the gas ability to eradicate the biofilm?
Response: Authors wish to thank reviewer 2 for his/her valuable comment. Yes, ceramic tiles were chosen due to the lack of scientific literature related to the formation of biofilm of K. pneumoniae on them. They were selected according to surface characteristics (roughness, hydrophilic). Physico-chemical characteristics of hydrophilic nature or roughness has direct impact on bacterial adhesion and subsequent biofilm formation on tiles. It is assumed that if the surface of the tile is rougher, the biofilm would be more difficult to remove. Bacterial adhesion to the surface is a very important step in the formation of biofilm, and the hydrophobicity of bacteria and the surface is dominant factor in the adhesion process. Katsikogianni and Missirlis found in their study that bacteria with hydrophobic characteristics prefer hydrophobic adhesion surfaces, which goes vice versa. So, if the tiles and bacteria correspond with hydrophilic nature (in our case), it is to be expected that bacteria will strongly adhere to the surface and produce high amount of biofilm. High amount of biofilm means more biomass for ozone to destroy. Also, papers like Megahed and Sharma reports that ozone efficacy is surface dependent.
Comment 5: Authors should spend more time to discuss the articles in which it is clearly shown that ozone completely destroy the germs at different doses and their selection of experimental variables (especially the fixed dosage 25 ppm).
Response: Authors wish to thank reviewer 2 for his/her valuable comment. Authors have added lines in the discussion about articles that used same dosage of 25 ppm and other concentrations. The main difference between majority of available articles and subsequent data is that mostly all of them refers to the planktonic bacterial form which is more susceptible to ozone action than bacterial biofilm. Furthermore, majority of data available on ozone action against K. pneumoniae biofilm refers to ozonized water, so actually only available data for ozone action against biofilm is the data on food pathogens and some S. aureus and P. aeruginosa. Authors think that all major articles that refers to the action of ozone on biofilm that can be correlated to results from the paper are available in the discussion.

Round 2
Reviewer 1 Report
Comment 9: Line 167: “the samples were homogenized” How did authors homogenize the ceramic tiles.
Response: Authors wish to say that the method for growing and enumerating biofilm formed on ceramic tiles was first described by Ivanković T. et al. Authors have used the method with some modifications. In the line 167 “the samples were homogenized” was meant to mean that after the ozone treatment, ceramic tiles from were carefully removed from agar and washed out with sterile saline solution, placed in sterile tube containing 10 mL of sterile saline and sonicated at ultrasound bath (Bandelin-BactoSonic, Germany) at 40 kHz for 1 minute to enhance release of the adhered cells from the tiles. Afterwards, sonicated sterile saline with tiles was homogenized during 1 minute on vortex to enhance further and final detachment of remaining cells from the biofilm into the sterile saline. Then 200 µL of this biofilm enriched sterile saline was used to make ten-fold serial dilutions. Manuscript was changed in a way that word “the samples were homogenized using vortex and aliquot of 200 µL of solution was used to prepare ten-fold serial dilutions”.
Reviewer suggestion: Therefore, suggesting authors to cite “Ivanković T. et al.” as a reference. In addition, the description in the response is clearer than the revised manuscript. Recommend authors revised the description in the manuscript based on this response.
Comment 11: Section 3.5: did authors used ozone concentration above 25 ppm? Reduction of 2 log was not very effective and apparently higher concentrations were needed to achieve safer results.
Reviewer suggestion: For scientific merit, it is better to show the antibacterial results of higher ozone concentrations though they did not increase antibacterial effects significantly. It is pretty common that the higher concentrations of disinfectant do not enhance the antibacterial effect. Thus, it is good to know the effective limit of disinfectant. It is also better for this article to present that 25 ppm is the optimal concentration. Thus, all following tests all used this concentration.
Comment 13: Figure 9: only one letter “a” was marked for statistically significance.
Reviewer suggestion: Authors did not mention that Figure 9 has been revised to Figure 10. Commonly, different letters indicate statistical significance. Since there is a statistical significance control and treatment
Comment 14: Line 415: Results indicated ceramic tiles used for this study were hydrophilic. These confuse me since most ceramic tiles on wall are hydrophobic and water-proof. This is why ceramic tiles are widely used in bathrooms and toilets. If authors use the rough side of tile, it is against the daily practice since the exposed side of tiles are smooth.
Reviewer suggestion: Authors did not mention that new line number of revised manuscript. However, after reading carefully, the revised description is fine.
Comment 15: Did authors test other treatment time? It would be a more complete study to have different treatment time periods.
Reviewer suggestion: The same reason for Comment 11, it is better to show the antibacterial results of longer treatment time though they did not increase antibacterial effects significantly. It is good to know the effective limit of disinfectant for a complete study.
Author Response
Response to Reviewer #1 of the manuscript ID: ijerph-1695574 entitled: ″Disinfecting action of gaseous ozone on OXA-48-producing Klebsiella pneumoniae biofilm in vitro” by Kaća Piletić, Bruno Kovač, Marko Perčić, Jure Žigon, Dalibor Broznić, Ljerka Karleuša, Sanja Lučić Blagojević, Martina Oder and Ivana Gobin
Comment 9: Line 167: “the samples were homogenized” How did authors homogenize the ceramic tiles.
Response: Authors wish to say that the method for growing and enumerating biofilm formed on ceramic tiles was first described by Ivanković T. et al. Authors have used the method with some modifications. In the line 167 “the samples were homogenized” was meant to mean that after the ozone treatment, ceramic tiles from were carefully removed from agar and washed out with sterile saline solution, placed in sterile tube containing 10 mL of sterile saline and sonicated at ultrasound bath (Bandelin-BactoSonic, Germany) at 40 kHz for 1 minute to enhance release of the adhered cells from the tiles. Afterwards, sonicated sterile saline with tiles was homogenized during 1 minute on vortex to enhance further and final detachment of remaining cells from the biofilm into the sterile saline. Then 200 µL of this biofilm enriched sterile saline was used to make ten-fold serial dilutions. Manuscript was changed in a way that word “the samples were homogenized using vortex and aliquot of 200 µL of solution was used to prepare ten-fold serial dilutions”.
Reviewer suggestion: Therefore, suggesting authors to cite “Ivanković T. et al.” as a reference. In addition, the description in the response is clearer than the revised manuscript. Recommend authors revised the description in the manuscript based on this response.
Response: Authors wish to thank to the reviewer #1 for his/her valuable comments. Authors wish to say that they did cite Ivanković T. et al in the references under number 39 – line 150 in the revised text and line 717-718. Also, authors wish to claim that manuscript was revised in a was that they added description parts into the sections 2.5 and 2.6 regarding clearer methodology.
Comment 11: Section 3.5: did authors used ozone concentration above 25 ppm? Reduction of 2 log was not very effective and apparently higher concentrations were needed to achieve safer results.
Reviewer suggestion: For scientific merit, it is better to show the antibacterial results of higher ozone concentrations though they did not increase antibacterial effects significantly. It is pretty common that the higher concentrations of disinfectant do not enhance the antibacterial effect. Thus, it is good to know the effective limit of disinfectant. It is also better for this article to present that 25 ppm is the optimal concentration. Thus, all following tests all used this concentration.
Response: Authors wish to to thank to the reviewer #1 for his/her valuable comments. Authors apologize for the fact that they didn't answer to the reviewer that original manuscript was revised in a way that Table 1. which was results from MATS test was deleted because of the result excess and transfiguered into text. New Figure 2A. was added with ozone reduction rates during different set of concentrations (lines 307-308 in the revised text). Also, authors wish to say that they added Figure 2B showing data with second exposure time was added in the Figure 2. Furthermore, authors did revise the manuscript in a way that they changed terminology of „minimum effective concentration“ to „optimal“ concentration and that manuscript was changed a bit to fit the new terminomogy and purpose.
Comment 13: Figure 9: only one letter “a” was marked for statistically significance.
Reviewer suggestion: Authors did not mention that Figure 9 has been revised to Figure 10. Commonly, different letters indicate statistical significance. Since there is a statistical significance control and treatment
Response: Authors wish to thank reviewer #1 for his/her valuable comment and apologize for the misinterpretation. Manuscript was revised in a way that Figure 10. has been amended with second „a“ fot the control. Also, as mentioned before, because of the data excess, authors decided to remove Table 1. from the original manuscript and transformulate the data from the MATS test info wording, rather then into table. Because of that and for scientific merrit, authors decided to add aditional Figure 2. with reduction rate with different ozone concentrations, so all other figures in the text changed their numbers.
Comment 14: Line 415: Results indicated ceramic tiles used for this study were hydrophilic. These confuse me since most ceramic tiles on wall are hydrophobic and water-proof. This is why ceramic tiles are widely used in bathrooms and toilets. If authors use the rough side of tile, it is against the daily practice since the exposed side of tiles are smooth.
Reviewer suggestion: Authors did not mention that new line number of revised manuscript. However, after reading carefully, the revised description is fine.
Response: Authors wish to thank reviewer #1 fot the comment and apologize for the mistake. As described before, Table 1. was removed from the original text and Figure 2. was added, so all other Figures changed their numbers.
Comment 15: Did authors test other treatment time? It would be a more complete study to have different treatment time periods.
Reviewer suggestion: The same reason for Comment 11, it is better to show the antibacterial results of longer treatment time though they did not increase antibacterial effects significantly. It is good to know the effective limit of disinfectant for a complete study.
Response: Authors wish to to thank to the reviewer #1 for his/her valuable comments. As mentioned before, authors wish to say that they revise the manuscript in a way that second exposure time was added in the Figure 2B. in the result section to better depict limits of disinfectant.
Reviewer 2 Report
The Authors apported the suggested correction.
Author Response
Authors wish to thank to the reviewer #2 for his/her valuable comments.

This manuscript is a resubmission of an earlier submission. The following is a list of the peer review reports and author responses from that submission.
Round 1
Reviewer 1 Report
In this manuscript, Piletić et al determined the effect of gaseous ozone on OXA-48 carbapenemase producing Klebsiella pneumoniae biofilm formation and bacterial survival.
Major comments:
- Table 1, The authors should describe why they only performed MIC on some strains/drugs.
- The authors found ozone treatment with 25 ppm for 1 hour exposure could reduce the bacterial number, therefore, I think that is the major reason why ozone treatment can reduce biofilm formation.
- Fig. 5. Which strain was used for this assay ? All 7 strains showed similar results ?
- Fig. 4. Did authors performed atomic force microscopy observation on all 7 strains ? They should offer the quantitative results. In addition, line 420, "Topology 419 of biofilm created after 24 hours (A, C and D)", I think it should be (A, B, and C) ?
- Fig. 7. Which strain was used for this assay ? Did authors observed the morphological changes of all 7 strains after ozone treatment ?
- Fig. 8.The authors should offer the quantitative results.
- Discussion. No need to repetitively described the results (e.g. lines 515-534). Therefore, the discussion section should be reorganized.
Minor comments:
- Line 79, put a space before "90%"
- "In vitro" should be written in italic, e.g. lines 99, 105, and 141
- Please use the abbreviation of K. pneumoniae (e.g. lines 136-140) since the full name of Klebsiella pneumoniae has been described in line 44.
- Line 230. "Crystal-violet staining" should not be capitalized. In addition, delete "Crystal-violet staining" in line 280.
- Line 338. I think "6.0 log10 to 6.5 0 log10" should be "6.0 log10 to 6.5 log10".
- Line 381. delete (Kp)
- Line 461, delete "reactive oxygen species"
Author Response
Response to Reviewer #1 of the manuscript ID: microorganisms-1595797 entitled: ″ Adverse effect of gaseous ozone on OXA-48 carbapenemase producing Klebsiella pneumoniae biofilm” by Kaća Piletić, Bruno Kovač, Marko Perčić, Jure Žigon, Dalibor Broznić, Ljerka Karleuša, Sanja Lučić Blagojević, Martina Oder and Ivana Gobin
General response:
The Authors would like to thank Reviewer #1 for her/his valuable comments on our manuscript. We have responded to all of the comments and revised the paper accordingly. Detailed responses to each comment are as follows.
Comment 1: Table 1. The authors should describe why they only performed MIC on some strains/drugs.
The authors found ozone treatment with 25 ppm for 1 hour exposure could reduce the bacterial number, therefore, I think that is the major reason why ozone treatment can reduce biofilm formation.
Response: In order to confirm multidrug resistant profile of the chosen standard and clinical bacterial strains used in this study, authors have determined antimicrobial profile of all 7 strains of K. pneumoniae. Table 1. was amended in a way that it was removed from the text and moved to the supplement data. Table 1. was also amended in a way that authors have performed analyses which was left out in Table 1. and have chosen to depict MIC only for colistin. MIC for colistin was depicted because colistin is drug of choice for carbapenemase producing OXA-48 K. pneumoniae and lately, some carbapenemase OXA-48 K. pneumoniae strains exhibit resistance towards colistin, therefore, the identification of colistin-resistance K. pneumoniae (Co-CCRKp) is urgent for fighting hospital acquired infections.
Comment 2: Fig. 5. Which strain was used for this assay ? All 7 strains showed similar results ?
Response: Digital microscopy was performed in triplicate on representative strain K. pneumoniae NCTC OXA 13442 since all 7 tested strains previously showed similar response to the ozone treatment. Authors decided to perform all assays regarding ozone mechanism of action on standard strain also for the reason of reasonable and sustainable funding of these delicate analyses.
Comment 3: Fig. 4. Did authors performed atomic force microscopy observation on all 7 strains ? They should offer the quantitative results. In addition, line 420, "Topology 419 of biofilm created after 24 hours (A, C and D)", I think it should be (A, B, and C) ?
Response: AFM was performed in triplicate on representative strain K. pneumoniae NCTC OXA 13442 since all 7 tested strains previously showed similar response to the ozone treatment. Authors decided to perform all assays regarding ozone mechanism of action on standard strain K. pneumoniae NCTC OXA 13442 also for the reason of economical limitations and for the reasonable and sustainable funding of these delicate analyses. In addition, Line 420 was amended in a way that in the brackets is written (A, B and C).
Comment 4: Fig. 7. Which strain was used for this assay ? Did authors observed the morphological changes of all 7 strains after ozone treatment ?
Response: SEM analyses were done on all 7 strains of K. pneumoniae (both clinical and standard strains) and all 7 strains showed similar results regarding morphological changes in biofilm structure and individual bacterial cells, so it was chosen to display and present SEM results from representative strain of K. pneumoniae NCTC OXA 13442 in the paper. Authors think that repetition of similar images of 7 strains would unnecessarily burden the length and complexity of the paper.
Comment 5: Fig. 8.The authors should offer the quantitative results.
Response: The authors would like to thank Reviewer #1 for her/his valuable comments on our manuscript and apologize for this oversight. Results are changed accordingly and quantitative results were added to this section, as well as one magnified picture where dead cells can be seen more easily.
Comment 6: Discussion. No need to repetitively described the results (e.g. lines 515-534). Therefore, the discussion section should be reorganized.
Response: Discussion was reorganized in a way that repetitive description of results was left out. Also, some additional text on ozone efficacy as solitary disinfectant was added regarding impossibility of solitary disinfectant to completely eradicate biofilm from inanimate surfaces (marked in track changes).
Comment 7: Line 79, put a space before "90%"
Response: Spacing was put before “90%” in line 79.
Comment 8: "In vitro" should be written in italic, e.g. lines 99, 105, and 141
Response: Wording “in vitro” has been changed to italic in lines 99, 105, 141 and throughout the text appropriately.
Comment 9: Please use the abbreviation of K. pneumoniae (e.g. lines 136-140) since the full name of Klebsiella pneumoniae has been described in line 44.
Response: Abbreviation of K. pneumoniae is used throughout the text (lines 136-140).
Comment 10: Line 230. "Crystal-violet staining" should not be capitalized. In addition, delete "Crystal-violet staining" in line 280.
Response: In line 230 capital letter in crystal – violet staining was corrected and wording “crystal – violet staining” was deleted from line 280.
Comment 11: Line 338. I think "6.0 log10 to 6.5 0 log10" should be "6.0 log10 to 6.5 log10".
Response: Text in line 338 was changed to 6.0 log10 to 6.5 log10.
Comment 12: Line 381. delete (Kp)
Response: (Kp) was deleted from the line 381.
Comment 13: Line 461, delete "reactive oxygen species"
Response: From the line 461 wording “reactive oxygen species” was deleted.

Reviewer 2 Report
This study aimed to investigate the effect of gaseous ozone on early K. pneumoniae biofilm in vitro. 7 strains of K. pneumoniae were used to form biofilm on ceramic tiles. The 24-hour biofilm of K. pneumoniae was exposed to 25 ppm of ozone gas for 1 hour in the experimental chamber.
Ozone gas in a concentration of 25 ppm was not sufficient to completely remove K. pneumoniae biofilm but certainly affected causing biofilm reduction and inhibition.
Ozone gas in a concentration of 25 ppm for 1 hour exposure time was not sufficient to remove early K. pneumoniae biofilm but did have a significant effect on several viable bacteria and total bacterial load. Morphological changes of biofilm topology and cell wall damage were observed in form of invaginations, as well as an increase of the intracellular level of reactive oxygen species (ROS). These results indicate that ozone has a good potential for hospital disinfection when used in combination with mechanical cleaning and combination with other disinfectants.
The study deals with a very sensitive issue concerning the presence of microorganisms in the hospital environment that can resist, through biofilms, the normal eradication and disinfection activities implemented by hospital sanitation protocols.
In particular, gaseous ozone has shown partial success in eradicating pathogens, although not in single-use, but probably needs to be combined with other types of eradicating agents.
If possible, I would highlight the hospital expenditure related to the use of this agent, to assess how optimal its use is, or whether it would be more useful to use other agents that might be more effective even in a single-use.
The precision and meticulousness with which the molecular mechanism and chemical composition of the disinfecting agent is described are truly remarkable.
I recommend highlighting and emphasizing how widespread the presence of the micro-organisms most sensitive to ozone gas is at present, since, in the pandemic period, we are witnessing a disappearance of the more resistant micro-organisms frequently isolated in the various departments.
In support of this I would recommend reading and if possible quoting the following articles:
PMID: 33498701
PMID: 33031863
With the following modifications, I consider the article publishable.
Author Response
Response to Reviewer #2 of the manuscript ID: microorganisms-1595797 entitled: ″ Adverse effect of gaseous ozone on OXA-48 carbapenemase producing Klebsiella pneumoniae biofilm” by Kaća Piletić, Bruno Kovač, Marko Perčić, Jure Žigon, Dalibor Broznić, Ljerka Karleuša, Sanja Lučić Blagojević, Martina Oder and Ivana Gobin
General comment:
This study aimed to investigate the effect of gaseous ozone on early K. pneumoniae biofilm in vitro. 7 strains of K. pneumoniae were used to form biofilm on ceramic tiles. The 24-hour biofilm of K. pneumoniae was exposed to 25 ppm of ozone gas for 1 hour in the experimental chamber. Ozone gas in a concentration of 25 ppm was not sufficient to completely remove K. pneumoniae biofilm but certainly affected causing biofilm reduction and inhibition. Ozone gas in a concentration of 25 ppm for 1 hour exposure time was not sufficient to remove early K. pneumoniae biofilm but did have a significant effect on several viable bacteria and total bacterial load. Morphological changes of biofilm topology and cell wall damage were observed in form of invaginations, as well as an increase of the intracellular level of reactive oxygen species (ROS). These results indicate that ozone has a good potential for hospital disinfection when used in combination with mechanical cleaning and combination with other disinfectants. The study deals with a very sensitive issue concerning the presence of microorganisms in the hospital environment that can resist, through biofilms, the normal eradication and disinfection activities implemented by hospital sanitation protocols.
In particular, gaseous ozone has shown partial success in eradicating pathogens, although not in single-use, but probably needs to be combined with other types of eradicating agents.
If possible, I would highlight the hospital expenditure related to the use of this agent, to assess how optimal its use is, or whether it would be more useful to use other agents that might be more effective even in a single-use. The precision and meticulousness with which the molecular mechanism and chemical composition of the disinfecting agent is described are truly remarkable.
I recommend highlighting and emphasizing how widespread the presence of the micro-organisms most sensitive to ozone gas is at present, since, in the pandemic period, we are witnessing a disappearance of the more resistant micro-organisms frequently isolated in the various departments. In support of this I would recommend reading and if possible quoting the following articles: PMID: 33498701 and PMID: 33031863. With the following modifications, I consider the article publishable.
General response: The Authors would like to thank Reviewer #2 for her/his valuable comments on our manuscript. We have responded to all of the comments and revised the paper accordingly. Detailed responses to each comment are provided below.
Comment 1: If possible, I would highlight the hospital expenditure related to the use of this agent, to assess how optimal its use is, or whether it would be more useful to use other agents that might be more effective even in a single-use.
Response: Authors would like to thank Reviewer #2 for her/his valuable comment and claim that they have made changes in Discussion relating hospital cost-effectiveness on ozone usage, not with exact costs of ozone usage because it varies from state to state, but in the sense of its comparison to chlorine based disinfectants (frequently used in hospital disinfection) and certain limitation but also the ways this limitations can be dealt with.
Comment 2: I recommend highlighting and emphasizing how widespread the presence of the micro-organisms most sensitive to ozone gas is at present, since, in the pandemic period, we are witnessing a disappearance of the more resistant micro-organisms frequently isolated in the various departments. In support of this I would recommend reading and if possible quoting the following articles: PMID: 33498701 and PMID: 33031863.
Response: Authors would like to thank Reviewer #2 for her/his valuable comment and valuable information and claim that they have made changes in Introduction and Discussion adding reference to following articles PMID: 33498701 and PMID: 33031863 and emphasising influence of more frequent cleaning/disinfecting strategies in hospital wards during COVID-19 pandemic, but also emphasising the fact that K. pneumoniae contamination in COVID-19 wards is on the rise despite more frequent cleaning, which was very interestingly put in this study by

Reviewer 3 Report
The manuscript aims to explore the effect of ozone on early K.pneumoniae biofilm, leading to the conclusion that such treatment is largely ineffective, thereby limiting its use as the late step after "traditional" cleaning/disinfection, which does not address the issue of the need to reach "hidden" bacteria in the nosocomial environment.
The introduction and the discussion provide useful information. However, the manuscript organization is quite disappointing, mainly because it contains much information poorly related to the claimed aims. In particular, the use of carbapenemase-producing strains appears poorly relevant (it is not expected difference in the sensitivity to ozone, related to antibiotic resistance), as well as strains characterization and information about antimicrobial-resistance profiles.
As for the style, various results subsections sound like M&M.
Despite the overall information about the poor viability of disinfection using ozone in the hospital environment, in my opinion, the manuscript is not worthy of publication in the Journal.
Author Response
Response to Reviewer #3 of the manuscript ID: microorganisms-1595797 entitled: ″ Adverse effect of gaseous ozone on OXA-48 carbapenemase producing Klebsiella pneumoniae biofilm” by Kaća Piletić, Bruno Kovač, Marko Perčić, Jure Žigon, Dalibor Broznić, Ljerka Karleuša, Sanja Lučić Blagojević, Martina Oder and Ivana Gobin
General comment:
The manuscript aims to explore the effect of ozone on early K.pneumoniae biofilm, leading to the conclusion that such treatment is largely ineffective, thereby limiting its use as the late step after "traditional" cleaning/disinfection, which does not address the issue of the need to reach "hidden" bacteria in the nosocomial environment.
The introduction and the discussion provide useful information. However, the manuscript organization is quite disappointing, mainly because it contains much information poorly related to the claimed aims. In particular, the use of carbapenemase-producing strains appears poorly relevant (it is not expected difference in the sensitivity to ozone, related to antibiotic resistance), as well as strains characterization and information about antimicrobial-resistance profiles.
As for the style, various results subsections sound like M&M.
Despite the overall information about the poor viability of disinfection using ozone in the hospital environment, in my opinion, the manuscript is not worthy of publication in the Journal.
General response: The Authors would like to thank Reviewer #3 for her/his valuable comments on our manuscript. We have responded to all of the comments and revised the paper accordingly. Detailed responses to each comment are provided below.
Comment 1: However, the manuscript organization is quite disappointing, mainly because it contains much information poorly related to the claimed aims. In particular, the use of carbapenemase-producing strains appears poorly relevant (it is not expected difference in the sensitivity to ozone, related to antibiotic resistance), as well as strains characterization and information about antimicrobial-resistance profiles.
Response: The authors would like to thank to Reviewer #3 for this comment and wish to add that results regarding antimicrobial profile of tested K. pneumoniae strains were removed from the main text and added to the supplement data in form of a Table. Also, authors would kindly like to add that carbapenemase producing K. pneumoniae – OXA 48 was chosen as target microorganism because of it’s predominance in hospital wards and it’s clinical significance, as well as for the fact that in times of rigorous disinfection with classical disinfectants in time of COVID-19, K. pneumoniae seems to endure (Bentivegna et al., 2021.) Furthermore, numerous previous studies on MDRO showed that it is more likely that MDRO organism are resistant to biocidal action of classical disinfections. To fully determine MDRO nature of our OXA-48 strains, authors have performed antimicrobial profile.
Authors would also like to emphasise that Table 2. – inhibition rates were moved from the main text to supplement data accordingly.
Comment 2: various results subsections sound like M&M
Response: Authors would like to thank Reviewer #3 for this comment and add that some Results sections were reorganised in a was to leave out paragraphs that describe used methods and repetitive explanations.
Comment 3: Despite the overall information about the poor viability of disinfection using ozone in the hospital environment…
Response: Authors would like to thank Reviewer #3 for this comment and wish to apologize for the eventual misconceptions regarding their overall presentation of ozone efficacy on K. pneumoniae biofilm. Ozone gas in concentration of 25 ppm during 1 hour exposure time caused 2.0 log10 to 2.5 log10 reduction in total number of cultivable bacteria. This reduction when calculated as inhibition rate varies from 97.8% to 99.36% in total. Furthermore, authors would like to emphasise that they have treated K. pneumoniae biofilm with ozone. Biofilm is known to be impossible to completely remove from inanimate surface with only one disinfectant, regardless of the method used and numerous studies claim persistence of MDRO pathogens biofilms in hospital dry surfaces after terminal disinfection. Majority hospital cleaning protocols and intensive preventive and control measures regarding hospital infections propose multiple sanitation methods always combined with mechanical cleaning. In this study, authors claimed that ozone gas has good potential to be used in hospitals when used in combination because it exhibits strong oxidative action on bacterial cells in biofilm and causes morphological changes in bacteria which puts them in state of oxidative stress and makes them susceptible to action of usually used biocides. Authors think that ozone gas can be successfully used in hospitals to treat MDRO biofilm contamination, but having in mind biofilm resistance characteristics, always in combination with mechanical cleaning and

Reviewer 4 Report
Piletić et al investigated the effect of gaseous ozone on the biofilm formation in Klebsiella pneumoniae. The methods are appropriate and the results are interesting.
I have some comments to be improved.
- Line 166 and 268. The “OXA-48” was used as a strain name in this manuscript, but it is a name of enzyme. We usually use “OXA-48-producing pneumoniae”. And also, the authors use “OXA - 48 14, 15, 16, 33 and 34” to name the strains, but it is easy to confuse with the β-lactamase.
- Line 266. NCTC 13442 is a specific strain name, it’s not appropriate to use “OXA NCTC 13442” to call the strain.
- Line 899. The abbreviation, VBNC, should be spell out for the first time. The authors mentioned that some reports used lower concentration of ozone gas to inhibit bacteria growth like 2 and 12 ppm. I suggest adding some information of choosing 25 ppm as the condition in this study. Or the authors can add the pre-test results with different concentrations of ozone gas.
- The line number are not continuous (line 470 to 497 and 922 to 976) and the first sentence of section 3.2 is not complete in line 497.
Author Response
Response to Reviewer #4
General comments:
Piletić et al investigated the effect of gaseous ozone on the biofilm formation in Klebsiella pneumoniae. The methods are appropriate and the results are interesting.
Response:
The Authors would like to thank Reviewer #4 for her/his valuable comments on our manuscript. We have responded to the comments and revised the paper accordingly.
Comment 1:
Line 166 and 268. The “OXA-48” was used as a strain name in this manuscript, but it is a name of enzyme. We usually use “OXA-48-producing pneumoniae”. And also, the authors use “OXA - 48 14, 15, 16, 33 and 34” to name the strains, but it is easy to confuse with the β-lactamase.
Response:
The Reviewer’s suggestions have been accepted. You are right that the name of the bacterium is OXA-48-producing K. pneumoniae and we have corrected this in the text. It is clear to us that the name of the strain could be confused with producing β-lactamase, so we explained this in the text and Figures.
Comment 2:
Line 266. NCTC 13442 is a specific strain name, it’s not appropriate to use “OXA NCTC 13442” to call the strain.
Response:
The Reviewer’s suggestions have been accepted and changed in the text.
Comment 3:
Line 899. The abbreviation, VBNC, should be spell out for the first time. The authors mentioned that some reports used lower concentration of ozone gas to inhibit bacteria growth like 2 and 12 ppm. I suggest adding some information of choosing 25 ppm as the condition in this study. Or the authors can add the pre-test results with different concentrations of ozone gas.
Response:
The Reviewer’s suggestions have been accepted and changed in the text. Lower ozone concentrations were tested for planktonic forms of bacteria. In our experiments, prior to the main experiment, we have conducted a pilot phase of the experiment in which we have used different concentrations and times of ozone exposure, starting with 5 ppm/30 minutes. We have gradually increased ozone concentration and exposure time until we have reached concentration which gave reproducible results – 25 ppm/ 1 hour exposure time. The results obtained with lower ozone concentrations were not repeatable and we had large oscillations. This is the reason why we have chosen and used this concentration.
Comment 4:
The line number are not continuous (line 470 to 497 and 922 to 976) and the first sentence of section 3.2 is not complete in line 497.
Response:
We do not see any inconsistencies in line number numbering in the Microsoft word document of manuscript.

Reviewer 5 Report
In this research paper the authors investigate the effect of one peculiar dose of ozone, 25 ppm for 1h, on various strains of Klebsiella pneumoniae that previously formed biofilms on ceramic tiles. They observed a reduction of the biomass up to 90% by 9 different technics. Overall, the science is correct, but I have some major and minor concerns:
Major comments:
- First, the language needs to be revised. It is often odd or confusing.
- The introduction is too long and present data that are not used in this research paper (for example, when one speaks about regulator genes, lectors expect to see mutants of these genes in the result section). It should be shortened to less than a page.
- The methods are often written with too much information. Though I usually appreciate very descriptive methods, here it is a bit too much and the obvious need to be shorten.
- The result section is oddly presented. It is separated by experiments rather than by arguments. The tittle of each section is the name of the experiment, where it should be the conclusion of the argument made on the section. The point that the author made is clear but is very redundant over 10 pages. It would have been nice if the author tested several doses of ozone, and/or ozone with other cleaning chemicals, as they discuss in their discussion section.
- As for the introduction, the discussion is too long and feels more like a review rather than a discussion of the results. It is fine to make parallels with other species and conditions, but 4 pages of discussions about the reduction of biomass by 25 ppm of ozone is too long and should be shorten to a maximum of 1.5 pages. The structure of the discussion is a bit off. The first page (lines 801 to 814) is an introduction, then I do not understand the point made lines 869 to 872 and lines 878 to 891. Then lines 900 to 907, the authors seem to discuss results they did not described in the result section. Same lines 1003 to 1011. Finally, everything after line 1106 is rather confusing and do not bring anything to this paper.
Minor comments:
- Line 46: Gram-negative
- Line 101: These virulence
- Line 113: such as
- Line 115 The major …
- Lines 209 and 211, n=5 should be spelled in full words
- Lines 471 to 497 do not exist
- Lines 517 and 519: units are missing. Also, 25log10?
- Figure 2: y axis should be cut for better appreciation of the results.
- Line 899: what does VBNC stands for?
- Line 1026: what does EPS stands for?
- Table S2: the line titles do not match the legend.
Author Response
Response to Reviewer #5
In this research paper the authors investigate the effect of one peculiar dose of ozone, 25 ppm for 1h, on various strains of Klebsiella pneumoniae that previously formed biofilms on ceramic tiles. They observed a reduction of the biomass up to 90% by 9 different technics. Overall, the science is correct, but I have some major and minor concerns:
Response:
The Authors would like to thank Reviewer #5 for her/his generous comment on our manuscript.
Major comments:
Comment 1:
First, the language needs to be revised. It is often odd or confusing.
Response:
Thanks for the suggestion. The text has been changed and reviewed by a native English speaker.
Comment 2:
The introduction is too long and present data that are not used in this research paper (for example, when one speaks about regulator genes, lectors expect to see mutants of these genes in the result section). It should be shortened to less than a page.
Response:
Thanks for the suggestion. The text has been reorganized according to suggestions.
Comment 3:
The methods are often written with too much information. Though I usually appreciate very descriptive methods, here it is a bit too much and the obvious need to be shorten.
Response:
Thanks for suggestion. In accordance with the recommendation, we have shortened the materials and methods.
Comment 4:
The result section is oddly presented. It is separated by experiments rather than by arguments. The tittle of each section is the name of the experiment, where it should be the conclusion of the argument made on the section. The point that the author made is clear but is very redundant over 10 pages. It would have been nice if the author tested several doses of ozone, and/or ozone with other cleaning chemicals, as they discuss in their discussion section.
Response:
for suggestion. In our experiments, prior to the main experiment, we have conducted a pilot phase of the experiment in which we have used different concentrations and times of ozone exposure, starting with 5 ppm/30 minutes. We have gradually increased ozone concentration and exposure time until we have reached concentration which gave reproducible results – 25 ppm/ 1 hour exposure time. The results obtained with lower ozone concentrations were not repeatable and we had large oscillations. This is the reason why we have chosen and used this concentration.
Comment 5:
As for the introduction, the discussion is too long and feels more like a review rather than a discussion of the results. It is fine to make parallels with other species and conditions, but 4 pages of discussions about the reduction of biomass by 25 ppm of ozone is too long and should be shorten to a maximum of 1.5 pages. The structure of the discussion is a bit off. The first page (lines 801 to 814) is an introduction, then I do not understand the point made lines 869 to 872 and lines 878 to 891. Then lines 900 to 907, the authors seem to discuss results they did not described in the result section. Same lines 1003 to 1011. Finally, everything after line 1106 is rather confusing and do not bring anything to this paper.
Response:
In accordance with the recommendation, we have entirely shortened and reorganised discussion. Thanks for your suggestion.
Minor comments:
- Line 46: Gram-negative (The Reviewer’s suggestions have been changed in the text.)
- Line 101: These virulence (The Reviewer’s suggestions have been changed in the text.)
- Line 113: such as (The Reviewer’s suggestions have been changed in the text.)
- Line 115 The major … (The Reviewer’s suggestions have been changed in the text.)
- Lines 209 and 211, n=5 should be spelled in full words (The Reviewer’s suggestions have been changed in the text.)
- Lines 471 to 497 do not exist – (We do not see any inconsistencies in line number numbering in the Microsoft word document of manuscript.)
- Lines 517 and 519: units are missing. Also, 25log10? (The Reviewer’s suggestions have been changed in the text.)
- Figure 2: y axis should be cut for better appreciation of the results. (The Reviewer’s suggestions have been changed in the text.)
- Line 899: what does VBNC stands for? (The Reviewer’s suggestions have been changed in the text.)
- Line 1026: what does EPS stands for? (The Reviewer’s suggestions have been changed in the text.)
- Table S2: the line titles do not match the legend. (The Reviewer’s suggestions have been changed in the text.)

Round 2
Reviewer 1 Report
The authors have addressed all my concerns and therefore I support publication of this manuscript.
Author Response
Response to Reviewer #1
Comment:
The authors have addressed all my concerns and therefore I support publication of this manuscript.
Response:
We highly value your decision and thank you for the opportunity to revise the manuscript. Reviewer suggestions and remarks were quite helpful and were incorporated them into the revised paper.
We thank you for your efforts.

Reviewer 3 Report
Dear Authors,
while appreciating your efforts to improve the manuscript, my overall opinion about it is nearly unchanged.
Toward a practical perspective, as I wrote in my past comment, your results do not justify the use of ozone as a disinfectant against Kp. It is true that a reduction of 2 logs was observed, but it is also true that you suggest using it in combination with other chemicals, which strongly limits the advantage of using a gas to reach hidden, poorly accessible bacteria. Moreover, your data refer to early (i.e. immature, unstructured) biofilm, while in the real-world we often have to fight mature biofilms. Last but not least, you do not provide data supporting a disinfection strategy where ozone is combined with other chemicals.
In my opinion, your manuscript, once further improved, should be submitted to some lower-IF Journal
Author Response
Response to Reviewer #3
Comment:
Dear Authors,
while appreciating your efforts to improve the manuscript, my overall opinion about it is nearly unchanged.
Toward a practical perspective, as I wrote in my past comment, your results do not justify the use of ozone as a disinfectant against Kp. It is true that a reduction of 2 logs was observed, but it is also true that you suggest using it in combination with other chemicals, which strongly limits the advantage of using a gas to reach hidden, poorly accessible bacteria. Moreover, your data refer to early (i.e. immature, unstructured) biofilm, while in the real-world we often have to fight mature biofilms. Last but not least, you do not provide data supporting a disinfection strategy where ozone is combined with other chemicals.
In my opinion, your manuscript, once further improved, should be submitted to some lower-IF Journal
Response:
We respect your decision. Some of reviewer suggestions and remarks were quite helpful and were incorporated into the revised paper.
Also, authors wish to claim that they have chosen 24-hour biofilm because the paper also refers to the importance of nosocomial infections caused by K. pneumoniae biofilm and their prevention. Keeping in mind strict and frequent hospital sanitation protocols, authors considered a 24-hour biofilm to be the most appropriate for the purposes of this study.
We thank you for your efforts.

Reviewer 5 Report
I still think the tittle of each sections in the results are not appropriate section tittles. These should be conclusive and not indicative of the methods. Example: "3.1. Antimicrobial resistance profile of K. pneumoniae" should be more like "3.1 The strains of K. pneumoniae are mulltidrug resistant" and so on.
Good effort on the introduction and discussion sections.